# Catalytic growth in a shared enzyme pool ensures robust control of centrosome size

Deb Sankar Banerjee[1,2], Shiladitya Banerjee[1,3]*

[1]Department of Physics, Carnegie Mellon University, Pittsburgh, United States; [2]James Franck Institute, University of Chicago, Chicago, United States; [3]School of Physics, Georgia Institute of Technology, Atlanta, United States

## eLife Assessment

This **valuable** work suggests a new physical model of centrosome maturation: a catalytic growth model with a shared enzyme pool. The authors provide **compelling** evidence to show that the model is able to reproduce various experimental results such as centrosome size scaling with cell size and centrosome growth curves in *C. elegans*, and that the final centrosome size is more robust to differences in initial centrosome size. While direct experimental support for this theory is currently lacking, the authors propose concrete experiments that could distinguish their shared-enzyme model from previously proposed alternatives.

*For correspondence:
sbanerjee347@gatech.edu

**Competing interest:** The authors declare that no competing interests exist.

**Abstract** Accurate regulation of centrosome size is essential for ensuring error-free cell division, and dysregulation of centrosome size has been linked to various pathologies, including developmental defects and cancer. While a universally accepted model for centrosome size regulation is lacking, prior theoretical and experimental works suggest a centrosome growth model involving autocatalytic assembly of the pericentriolar material. Here, we show that the autocatalytic assembly model fails to explain the attainment of equal centrosome sizes, which is crucial for error-free cell division. Incorporating latest experimental findings into the molecular mechanisms governing centrosome assembly, we introduce a new quantitative theory for centrosome growth involving catalytic assembly within a shared pool of enzymes. Our model successfully achieves robust size equality between maturing centrosome pairs, mirroring cooperative growth dynamics observed in experiments. To validate our theoretical predictions, we compare them with available experimental data and demonstrate the broad applicability of the catalytic growth model across different organisms, which exhibit distinct growth dynamics and size scaling characteristics.

## Introduction

Centrosomes are membraneless organelles that act as microtubule organizing centers during mitotic spindle formation (*Gould and Borisy, 1977*). Prior to cell division, centrosomes grow many folds in size by accumulating various types of proteins including microtubule nucleators, in a process known as centrosome maturation (*Palazzo, 1999*). Tight control of centrosome size is functionally important for the cell as aberrations in centrosome growth and size can lead to errors in chromosome segregation (*Krämer et al., 2002*). This may result in aneuploidy, which is associated with a range of problems, including birth defects, developmental abnormalities, and cancer (*Basto et al., 2008*; *D'Assoro et al., 2002*; *Levine et al., 2017*). Previous works have suggested that centrosomes grow cooperatively and regulate their size through a coordinated assembly of the pericentriolar material, mediated by complex signaling pathways and regulatory proteins (*Alvarez Rodrigo et al., 2019*; *Conduit et al., 2014b*; *Zwicker et al., 2014*). Despite the significant progress on uncovering the molecular components

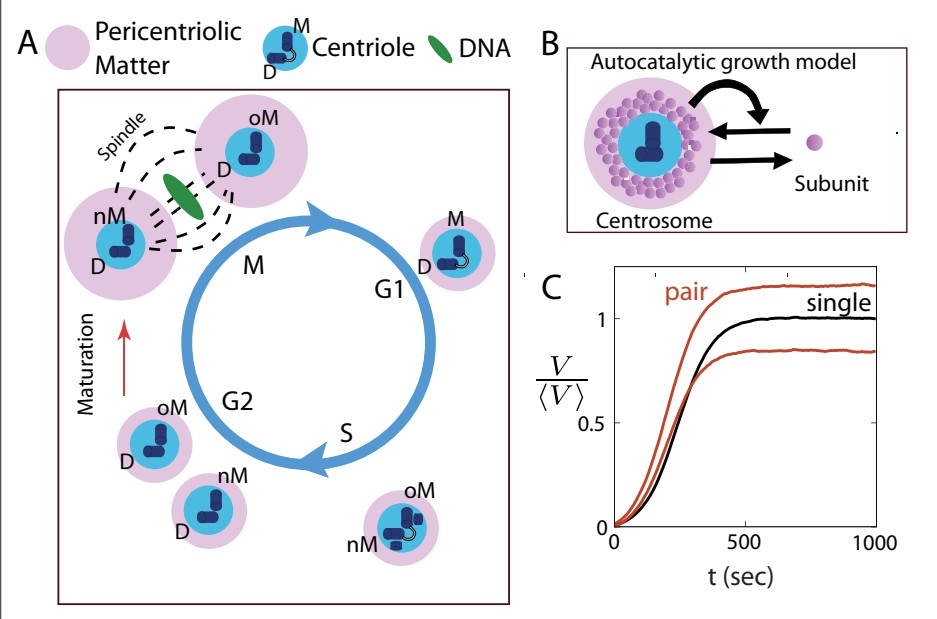

**Figure 1.** Autocatalytic feedback in centrosome growth drives centrosome size inequality. (**A**) Schematic showing the dynamics of centrosomes during the cell cycle. In the G1 phase, there is a single centrosome with mother (M) and daughter (D) centrioles at the core, surrounded by the pericentriolar material (PCM). The two new centriole pairs with the old mother (oM) and the new mother (nM) separate into two centrosomes in the G2/M phase after centriole duplication. The spatially separated centrosomes then grow via a process called *centrosome maturation* (red arrow), prior to cell division. (**B**) Schematic of the autocatalytic growth model for centrosomes, where the assembly rate increases with increasing centrosome size. (**C**) Autocatalytic growth of centrosomes captures the sigmoidal size dynamics for a single and a pair of centrosomes, but is unable to ensure size equality of a centrosome pair. See *Table 1* for a list of parameter values.

regulating centrosome assembly (*Conduit et al., 2015b*), a quantitative model connecting the molecular mechanisms of growth to centrosome size regulation is lacking.

Centrosomes are composed of a porous scaffold-like structure (*Schnackenberg et al., 1998*; *Feng et al., 2017*) known as the pericentriolar material (PCM), organized around a pair of centrioles at the core (*Figure 1A*). An individual cell starts with a single centrosome in the G1 phase, undergoes centriole duplication in the S phase, followed by the formation of two centrosomes in the G2/M phase (*Figure 1A*). During centrosome maturation, the two spatially separated centrosomes grow in size by adding material to their PCMs from a cytoplasmic pool of building blocks (*Decker et al., 2011*; *Woodruff et al., 2014*; *Kemp et al., 2004*; *Pelletier et al., 2004*; *Conduit et al., 2014b*), while the centrioles themselves do not grow. Following maturation, the two centrosomes achieve equal sizes (*Decker et al., 2011*; *Zwicker et al., 2014*; *Alvarez Rodrigo et al., 2019*), which is deemed essential in the establishment of a symmetric bipolar spindle (*Conduit et al., 2015b*). This size equality is vital for ensuring error-free cellular division, as spindle size is directly proportional to centrosome sizes (*Greenan et al., 2010*). However, the mechanisms by which centrosomes within a cell achieve equal size remain poorly understood.

A variety of qualitative and quantitative models of centrosome size regulation have emerged in recent years. These include the limiting pool theory (*Decker et al., 2011*; *Goehring and Hyman, 2012*), liquid-liquid phase separation model for PCM assembly (*Zwicker et al., 2014*), reaction-diffusion models (*Mahen et al., 2011*; *Mahen and Venkitaraman, 2012*), and centriole-driven assembly of PCM (*Conduit et al., 2010a*, *Conduit et al., 2014b*; *Alvarez Rodrigo et al., 2019*; *Kirkham et al., 2003*; *Banerjee and Banerjee, 2022*). While there is no universally accepted model for centrosome size regulation, all these models indicate a positive feedback mechanism underlying centrosome assembly. For instance, *Zwicker et al., 2014* described PCM assembly in *C. elegans* as an autocatalytic process, assembled from a single limiting component undergoing active phase segregation through centriole activity. The authors suggested that a modified version of this model may also

**Table 1.** Parameter Values.

| Figure | Parameter Values | Reference |
|---|---|---|
| *Figure 1C* | $\rho_0(=N/V_c)$=0.033 $\mu$M, $k_0^+ = 600\,\mu\mathrm{M}^{-1}\,s^{-1}$, $k_1^+ = 0.6\,\mu\mathrm{M}^{-1}\,s^{-1}$, $k^- = 0.005\,s^{-1}$ | based on *Zwicker et al., 2014* |
| *Figure 2A* | $\rho_0$=0.033 $\mu$M, $k^- = 0.005\,s^{-1}$ | based on *Zwicker et al., 2014* |
| *Figure 2D* | $k^+ = 1000\,\mu\mathrm{M}^{-1}\,s^{-1}$, $\rho_0$=0.1 $\mu$M | - |
| *Figure 3C* | $\rho_0 = 1\,\mu$M, $[E] = 0.1\,\mu$M, $k^+ = 1\,\mu\mathrm{M}^{-1}s^{-1}$, $k^* = 1000\,\mu\mathrm{M}^{-1}s^{-1}$, $k_E^* = 5\,s^{-1}$, $k_1^* = 1\,\mu\mathrm{M}^{-1}s^{-1}$ | - |
| *Figure 3D* | $\rho_0 = 0.05\,\mu$M, $[E] = 0.1\,\mu$M, $k^+ = 1\,\mu\mathrm{M}^{-1}s^{-1}$, $k^* = 2000\,\mu\mathrm{M}^{-1}s^{-1}$, $k_E^* = 10\,s^{-1}$, $k_1^* = 100\,\mu\mathrm{M}^{-1}s^{-1}$ | - |
| *Figure 3E* | $\rho_0 = 0.02\,\mu$M, $[E] = 0.09\,\mu$M, $k^+ = 1\,\mu\mathrm{M}^{-1}s^{-1}$, $k^* = 8 \times 10^4\,\mu\mathrm{M}^{-1}s^{-1}$, $k_E^* = 4.25\,s^{-1}$, $k_1^* = 0.1\,\mu\mathrm{M}^{-1}s^{-1}$ | From fitting experimental data |
| *Figure 3F* | $\rho_0 = 1\,\mu$M, $k^* = 1000\,\mu\mathrm{M}^{-1}s^{-1}$, $k_E^* = 5\,s^{-1}$, $k_1^* = 10\,\mu\mathrm{M}^{-1}s^{-1}$ | - |
| *Figure 3G & H* | $\rho_0 = 0.15\,\mu$M, $[E] = 0.085\,\mu$M, $k^+ = 1\,\mu\mathrm{M}^{-1}s^{-1}$, $k^* = 1000\,\mu\mathrm{M}^{-1}s^{-1}$, $k_E^* = 1\,s^{-1}$, $k_1^* = 5\,\mu\mathrm{M}^{-1}s^{-1}$ | - |
| *Figure 4A* | Same as *Figure 3E* | From fitting experimental data |
| *Figure 4B* | $\rho_0 = 0.1\,\mu$M, $[E] = 0.05\,\mu$M, $k^+ = 100\,\mu\mathrm{M}^{-1}s^{-1}$, $k^* = 2000\,\mu\mathrm{M}^{-1}s^{-1}$, $k_E^* = 10\,s^{-1}$, $k_1^* = 100\,\mu\mathrm{M}^{-1}s^{-1}$ | - |
| *Figure 4D* | $\rho_0 = 0.02\,\mu$M, $[E_{ss}^*] = 0.01\,\mu$M, $k_1^* = 0.1\,\mu\mathrm{M}^{-1}s^{-1}$, $V_c = 25000\,\mu m^3$ | - |
| *Figure 5B & D* | $\rho_0 = 0.5\,\mu$M, $[E] = 0.1\,\mu$M, $k^+ = 60\,\mu\mathrm{M}^{-1}s^{-1}$, $k^* = 2000\,\mu\mathrm{M}^{-1}s^{-1}$, $k_E^* = 10\,s^{-1}$, $k_1^* = 100\,\mu\mathrm{M}^{-1}s^{-1}$ | - |
| *Figure 5C & E* | $\rho_0$=0.033 $\mu$M, $k_0^+ = 60\,\mu\mathrm{M}^{-1}\,s^{-1}$, $k_1^+ = 0.6\,\mu\mathrm{M}^{-1}\,s^{-1}$, $k^- = 0.005\,s^{-1}$ | based on *Zwicker et al., 2014* |
| *Figure 6B* | $[\rho_a] = 0.25\,\mu$M, $[\rho_b] = 0.35\,\mu$M, $[\rho_E] = 0.015\,\mu$M, other parameters are same as below | - |
| *Figure 6C & D* | $[\rho_a] = 0.25\,\mu$M, $[\rho_b] = 0.5\,\mu$M, $[\rho_E] = 0.01\,\mu$M, $k_a^+ = 10\,\mu\mathrm{M}^{-1}s^{-1}$, $k_{b0}^+ = 0.5\,\mu\mathrm{M}^{-1}s^{-1}$, $k_{b0}^- = 0.01\,s^{-1}$, | - |
|  | $k_{aE}^+ = 5 \times 10^3\,\mu\mathrm{M}^{-1}s^{-1}$, $k_{Eb}^+ = 10^3\,\mu\mathrm{M}^{-1}s^{-1}$, $k_{b1}^+ = 10^4\,\mu\mathrm{M}^{-1}s^{-1}$, $k_{b1}^- = 5 \times 10^{-3}\,s^{-1}$, $k_a^- = 5 \times 10^{-3}\,s^{-1}$ |  |
| Fixed parameters | $\delta v = 2 \times 10^{-4}\,\mu m^3$, $V_0 = 5 \times 10^{-3}\,\mu m^3$, $k^- = 5 \times 10^{-3}s^{-1}$, $V_c = 5000\,\mu m^3$ | estimates & *Zwicker et al., 2014* |

apply to centrosome maturation in *Drosophila*. While this model captures sigmoidal growth dynamics observed experimentally and the scaling of centrosome size with cell size, autocatalytic growth of centrosome pairs can induce significant discrepancies in size. We discuss how small initial differences in centrosome size could be amplified during the process of autocatalytic growth, as the larger centrosome would incorporate more material, thereby outcompeting the smaller one.

Another category of models, based on a large body of recent experimental works on *Drosophila* (*Conduit et al., 2010a*, *Conduit et al., 2014b*; *Alvarez Rodrigo et al., 2019*; *Raff, 2019*), suggests that PCM assembly occurs locally around the centriole, driven by a positive feedback loop between the scaffold-former PCM components such as Centrosomin (Cnn) and Spindle defective-2 (Spd-2) facilitated by enzymes like Polo or Polo-like-kinase (Plks; *Alvarez Rodrigo et al., 2019*) and this mechanism of growth appears to remain conserved across different organisms enacted by functionally homologous proteins for example SPD-5 and SPD-2 in worms (*Alvarez Rodrigo et al., 2019*; *Raff, 2019*; *Aljiboury and Hehnly, 2023*).

In a recent study, we employed quantitative modeling to demonstrate that localized assembly around the centriole, accompanied by distributed turnover within the PCM, can ensure centrosome size equality (*Banerjee and Banerjee, 2022*). However, this model did not take into account positive feedback between PCM components, and was thus unable to capture the cooperative nature of growth dynamics. Thus, none of the existing quantitative models can account for robustness in

centrosome size equality in the presence of positive feedback. Furthermore, intracellular noise and the distinct nature of centrioles within the two centrosomes (old mother centriole and new mother centriole, depicted in *Figure 1A*) can give rise to fluctuations in centrosome size and introduce initial disparities in size during the maturation process. Consequently, a robust size regulation mechanism is required to achieve centrosome size parity, despite the presence of noise in growth and initial size differences.

Here, we present a quantitative theory for size regulation of a centrosome pair via catalytic assembly of the PCM from a cytoplasmic pool of enzymes and molecular components. We first establish that autocatalytic growth of centrosomes in a shared subunit pool results in amplification of initial size differences, leading to significant size inequality after maturation. Then we propose a new model of catalytic growth of centrosomes in a shared pool of building blocks and enzymes. Our theory is based on recent experiments uncovering the interactions of the molecular components of centrosome assembly, that is Polo-dependent positive feedback between Cnn and Spd-2 in *Drosophila* (*Conduit et al., 2010a*, *Conduit et al., 2014b*; *Alvarez Rodrigo et al., 2019*), and conserved functionally similar proteins that may constitute a similar pathway in other organisms like *C. elegans*, Xenopus, Zebrafish, and Human (*Raff, 2019*; *Aljiboury and Hehnly, 2023*). We show that this model ensures robust size control of centrosomes while capturing several key features of centrosome growth observed experimentally, including the growth of two stable centrosomes of equal size after maturation observed in *Drosophila* (*Conduit et al., 2015b*) and *C. elegans* (*Zwicker et al., 2014*), sigmoidal growth dynamics (*Zwicker et al., 2014*; *Decker et al., 2011*) and tunable scaling of centrosome size with cell size and centrosome number observed in *C. elegans* (*Decker et al., 2011*) and *Drosophila* (*Wong et al., 2022*), and the ability to robustly create centrosomes of different size from differences in centriole activity as observed in *Drosophila* male germ line stem cells (*Conduit and Raff, 2010b*) and larval neuroblasts (*Januschke et al., 2013*). We show that our model can explain seemingly different growth behaviours seen in worms and flies by comparing theoretical results with experimentally observed trends from these different organisms demonstrating the potential applicability of our model across different species. We further develop a two-component model of catalytic growth to explicitly show that without the sharing of the enzyme pool, centrosome size regulation is not robust when accounting for the experimentally observed enzyme-mediated positive feedback between the two components (*Alvarez Rodrigo et al., 2019*).

## Results

### Autocatalytic feedback in centrosome growth drives centrosome size inequality

Previous quantitative modeling of centrosome growth in *C. elegans* has suggested that centrosomes are autocatalytic droplets growing via phase separation in a limited pool of building blocks (*Zwicker et al., 2014*; *Decker et al., 2011*). Autocatalytic growth arises if the centrosome assembly rate increases with centrosome size, creating a size-dependent positive feedback (*Figure 1B*). To investigate if autocatalytic growth can ensure size equality of centrosomes, we considered a reaction-limited model of centrosome growth via stochastic assembly and disassembly of its subunits. Theoretical estimates indicate that the timescale of diffusion is much faster than the timescales of reactions observed in experiments. For instance, the scaffold formers diffuse over $5 - 10$ μm in about $1\,s$ while they have turnover timescale of $\sim 100\,\text{s}$ (see Materials and methods section for more details). Although there are multiple essential components involved in PCM assembly (*Dobbelaere et al., 2008*; *Conduit et al., 2010a*, *Conduit et al., 2014b*), we first examined a one-component centrosome model to illustrate the role of autocatalytic growth on size control. The deterministic description for the growth of a centrosome pair is given by

$$\frac{\mathrm{d}n_i}{\mathrm{d}t} = (k_0^+ + k_1^+ n_i(t))\rho(t) - k^- n_i(t) , \tag{1}$$

where $n_i(t)$ is the number of subunits in $i^{th}$ centrosome ($i = 1, 2$), $k_0^+$ and $k_1^+$ are the rate constants for non-cooperative and cooperative assembly, respectively, and $k^-$ is the disassembly rate constant. *Equation 1* can be derived from the phase segregation model for centrosome assembly studied by *Zwicker et al., 2014* (see Appendix), with $k_0^+$ and $k_1^+$ representing centriole activity and the strength of

autocatalytic interaction, respectively. In *Equation 1*, $\rho(t)$ is the cytoplasmic concentration of centrosomal subunits, given by $\rho(t) = (N - n_1(t) - n_2(t))/V_c$ where $V_c$ is cell volume and $N$ is the total amount of subunits in the cell. Centrosome volume is given by $V_i(t) = n_i(t)\delta v$, where $\delta v$ is the effective volume occupied by a single subunit. As shown before (*Zwicker et al., 2014*), this model can capture the essential quantitative features of the growth of a single centrosome (*Figure 1C*), including sigmoidal growth curve, temporal control of size and scaling of centrosome size with cell size. However, this model is unable to ensure the size equality of two identical centrosomes growing from a shared subunit pool. Stochastic simulation of this model, using the Gillespie algorithm (see Materials and methods), shows a significant difference in steady-state size even with a small initial size difference (*Figure 1C*).

It is instructive to first compare two opposite limits of the model, $k_0^+ = 0$ (purely autocatalytic growth) and $k_1^+ = 0$ (non-cooperative growth). For $k_0^+ = 0$, *Equation 1* can be interpreted as assembly and disassembly occurring throughout the PCM volume, with the assembly rate scaling with centrosome size. As a result, the centrosome with a larger initial size would end up growing to a larger steady-state size. Stochastic simulations of this model show that the ensemble-averaged absolute difference in centrosome size ($|\delta V| = |V_1 - V_2|$) increases with the initial centrosome size difference ($\delta V_0$), indicating lack of robustness in size regulation (see Appendix 1 and *Figure 2—figure supplement 1*). On the other hand, the limit $k_1^+ = 0$ corresponds to a model where the assembly rate is size-independent, and material turnover is distributed throughout the PCM volume. This model guarantees size equality of a centrosome pair competing for a limiting subunit pool (see Appendix 1 and *Figure 2—figure supplement 2*), even in the presence of large initial size differences (*Figure 2D*), with the steady-state size given by $V = k^+ N\delta v/(k^- + 2k^+)$. However, the resulting growth curve is non-sigmoidal, thus fails to capture experimental data in *C. elegans* (*Decker et al., 2011*; *Zwicker et al., 2014*).

To quantify the robustness of size control, we measured the relative difference in steady-state centrosome size, $|\delta V|/\langle V\rangle$, starting with an initial size difference $\delta V_0 \sim 0.01\langle V\rangle$, where $|...|$ denotes the absolute value and $\langle V\rangle$ is the ensemble average of centrosome size at steady-state. For a robust size regulation mechanism, the final size difference is expected to be independent of the initial size difference. The resulting size inequality is controlled by the rate constants $k_0^+$, $k_1^+$, $k^-$ and the pool size $N$. Our analysis shows that there is a relatively small region of the parameter space where the strength of the autocatalytic feedback is low enough to ensure a small difference in centrosome size (*Figure 2A*). Through linearization of the rate equations, we derive the analytical condition for size equality to be $2k_0^+ + k^- V_c > k_1^+ N$ (see Appendix 3 for details). However, in this range of parameter values, the growth is essentially non-cooperative and the growth curve is not sigmoidal (*Figure 2B*). Larger size inequality is associated with higher values of $k_1^+$, when the growth dynamics is sigmoidal in nature (*Figure 2C*). For a detailed study of the lack of robustness in size regulation, please refer to Appendix 1 and *Figure 2—figure supplement 3*.

While our theoretical estimates suggest that centrosome growth is primarily reaction-limited, the increasing distance between centrosomes during maturation — especially in certain organisms or depending on cell size—could lead to a diffusion-limited growth scenario. To investigate how diffusion affects centrosome size regulation, we extended our model to include subunit diffusion (see Appendix 4). Our results indicate that diffusion does not qualitatively alter centrosome size regulation. Size inequality can be reduced when the diffusion constant is low or when centrosomes are far apart, though in this regime, the growth curves lose their characteristic sigmoidal shape (see Appendix 4 and *Figure 2—figure supplement 4*). Crucially, the presence of diffusion does not resolve the issue of robustness in size control; the size difference between centrosomes still increases with larger initial size disparities (*Figure 2—figure supplement 4*).

## Catalytic growth in a shared enzyme pool ensures centrosome size equality and cooperative growth

### Model motivation and assumptions

Centrosome growth during maturation occurs through the expansion of a scaffold-like structure and subsequent recruitment of PCM proteins on the scaffold. While multiple proteins are involved in the scaffold assembly, Spd-2 and centrosomin (Cnn) are two essential scaffold-forming proteins identified in *Drosophila*, in the absence of which centrosome growth is almost entirely diminished (*Conduit et al., 2014a*). The kinase Polo interacts with both Spd-2 and Cnn to promote the assembly of a stable scaffold.

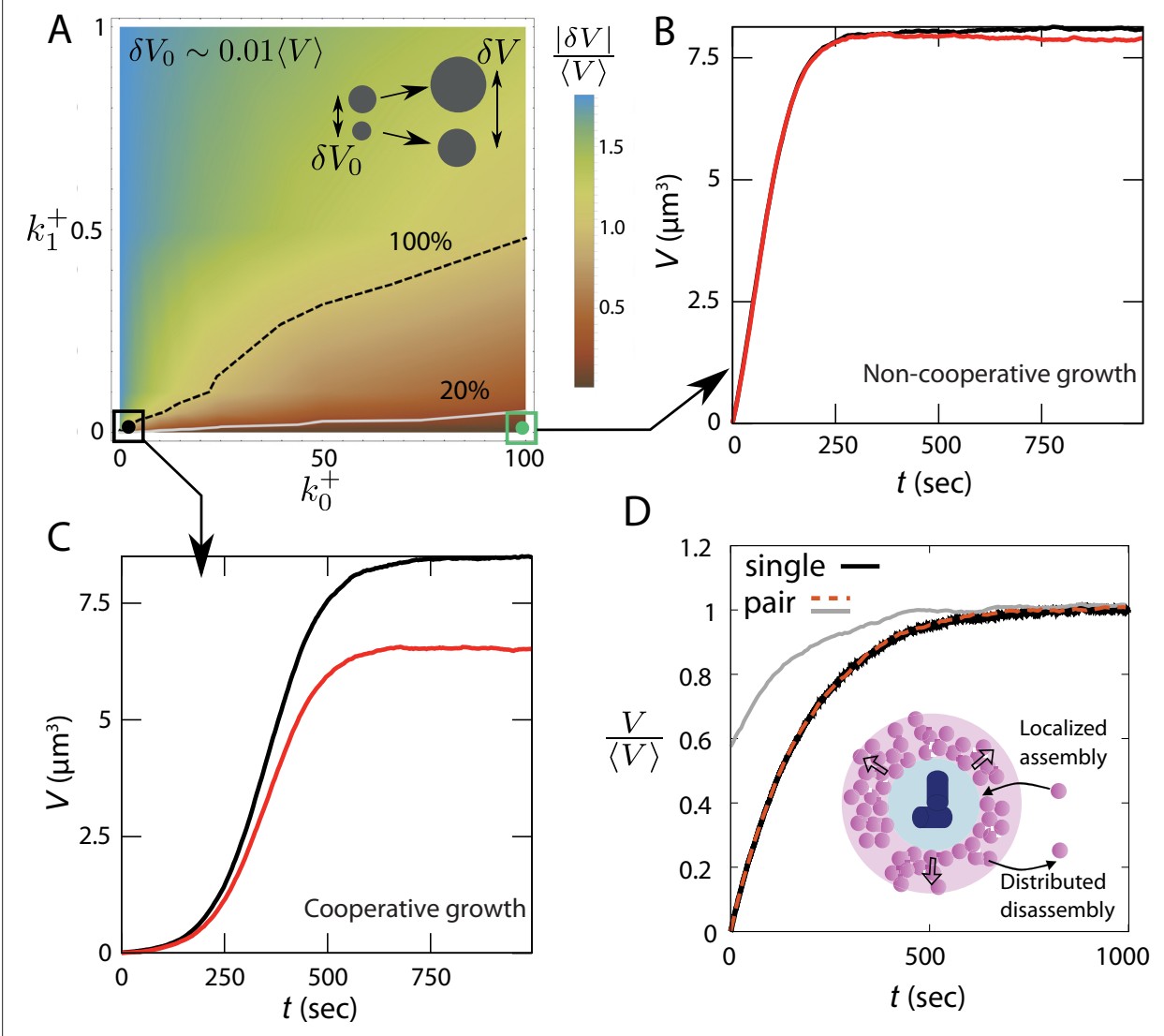

**Figure 2.** Lack of robust size control in autocatalytic growth. (**A**) The relative difference in centrosome size, $|\delta V|/\langle V\rangle$, as a function of the growth rate constants $k_0^+$ and $k_1^+$, with an initial size difference of $0.1\ \mu m^3$. The light gray and dashed black lines represent the lines $|\delta V|/\langle V\rangle = 0.2$ and $|\delta V|/\langle V\rangle = 1.0$. (**B,C**) Size dynamics of a pair centrosomes for (**B**) weakly cooperative ($k_0^+ = 100$, $k_1^+ = 0.001$) and (**C**) strongly cooperative ($k_0^+ = 0.1$, $k_1^+ = 0.001$) growth regimes. (**D**) Dynamics of centrosome size for a single centrosome and a pair of centrosomes simulated using the non-cooperative growth model. Inset: Schematic of centrosome growth via centriole-localized assembly and disassembly distributed throughout the PCM. The $|\delta V|/\langle V\rangle$ values in (**A**) represent an average over 1000 ensembles. The values of $k_0^+$ and $k_1^+$ are in the units of $\times 600\ \mu M^{-1}\ s^{-1}$. See **Table 1** for a list of parameter values. Parameter values for panel D were chosen to obtain typical steady-state centrosome size ($\sim 5\ \mu m^3$) and timescale of growth ($\sim 500$ s).

The online version of this article includes the following figure supplement(s) for figure 2:

**Figure supplement 1.** Failure of size regulation in purely autocatalytic growth.

**Figure supplement 2.** Growth via localized assembly and distributed disassembly.

**Figure supplement 3.** Centrosome size inequality and size dynamics in the autocatalytic growth model in different parameter regimes.

**Figure supplement 4.** Diffusion-limited growth mitigates centrosome size inequality but lacks sigmoidal nature.

In particular, Spd-2 recruits Cnn with the help of Polo and Cnn in turn strengthens the Spd-2 scaffold without directly recruiting additional Spd-2 proteins. Without the Polo kinase, the Cnn scaffold fails to grow (**Alvarez Rodrigo et al., 2019**). Similar molecular pathways exist in other organsisms like *C. elegans*, involving homologous proteins (**Raff, 2019**). These findings suggest a model for catalytic assembly of centrosomes based on positive feedback between scaffold-forming proteins and an enzyme. Moreover, Fluorescent Recovery After Photobleaching (FRAP) data reveal that the turnover rate of the enzyme Polo

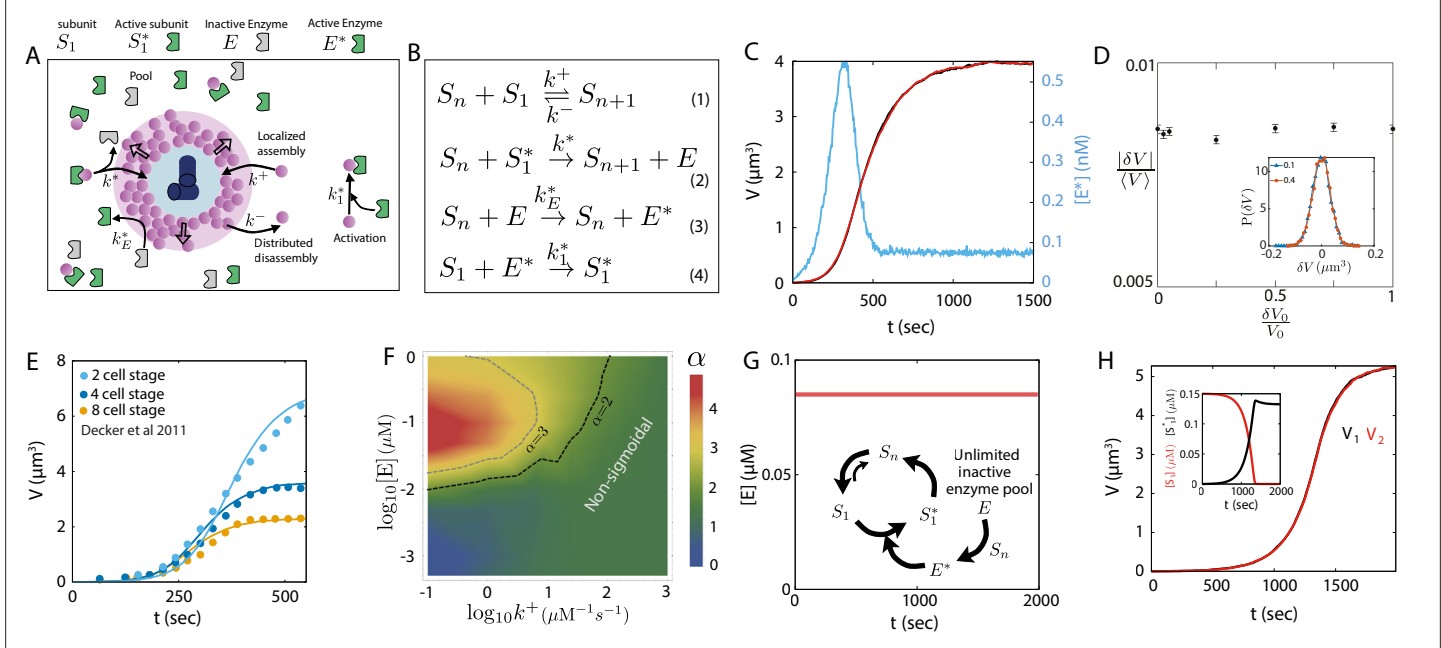

**Figure 3.** Catalytic growth in a shared enzyme pool leads to robust size control of a centrosome pair. (**A**) Schematic of centrosome growth via catalytic activity of an enzyme that is activated by PCM proteins at a rate proportional to PCM size. (**B**) Reactions describing centrosome growth via catalytic activity of enzyme $E$. The centrosome ($S_n$) can activate the enzyme in a state $E^*$, which in turn creates an activated subunit ($S_1^*$) that binds the PCM. (**C**) Size dynamics of a centrosome pair (blue, red curves) growing via catalytic assembly and the dynamics of the activated enzyme ($[E^*]$) in time (blue curve). (**D**) The ensemble average of relative absolute size difference $|\delta V|/\langle V \rangle$ is insensitive to change in relative initial size difference $\delta V_0/V_0$. Inset: Probability distribution of $\delta V$ for two different values of initial size difference ($\delta V_0/V_0 = 0.1$ and $\delta V_0/V_0 = 0.4$). (**E**) Centrosome growth curves obtained from the catalytic growth model (lines) fitted to experimental growth curves (points) measured at different stages of *C. elegans* development. (**F**) Degree of sigmoidal growth, measured by Hill coefficient $\alpha$, as a function of the growth rate constant $k^+$ and the total enzyme concentration $[E]$. (**G**) Model of shared catalysis considering a constant concentration of inactive enzyme ($E$) throughout the growth period. Inset: Schematic of the reactions showing the steady state cycle between $S_1$, $S_1^*$ and $S_n$. (**H**) Centrosome pair growth in the presence of unlimited inactive enzyme pool exhibits size equality as well as cooperative growth dynamics. Inset: Dynamics of $S_1$ and $S_1^*$ concentrations. See *Table 1* for a list of parameter values. Parameters were chosen to match typical steady-state centrosome size (~ 5 $\mu m^3$) and the timescale of growth (~ 500 $S$). Parameters for panel E were obtained by fitting the enzyme kinetics.

The online version of this article includes the following figure supplement(s) for figure 3:

**Figure supplement 1.** Origin and regulation of the enzyme pulse.

**Figure supplement 2.** Effect of pool size and correlation between final and initial size difference.

**Figure supplement 3.** Effect of subunit diffusion on catalytic growth.

kinase within PCM is much faster (~ 1 min) compared to the Spd-2 and Cnn (~ 10 min; *Alvarez Rodrigo et al., 2019*; *Wong et al., 2022*). Consequently, owing to the enzyme's pronounced diffusivity, there is a strong likelihood that the active enzyme pool is shared between the two centrosomes.

To determine whether a shared catalytic growth model can yield size parity in a pair of centrosomes, we initially formulated a single-component model for PCM growth, catalyzed by an enzyme (*Figure 3A*). This model takes into account a shared limiting pool of enzyme and PCM subunits. The assumption of a limiting subunit pool is supported by prior research on *C. elegans*, which displayed centrosome size scaling with centrosome number (*Decker et al., 2011*). While the presence of such a limited subunit pool has not been established in other systems, we will subsequently demonstrate that even in cases where centrosome size scaling is not pronounced, the subunit pool can still be finite. Consequently, we implement a model with a limiting pool for both subunits and enzymes. We later relax this assumption by exploring the implications of an infinite enzyme pool.

## Model description

In the single-component model for PCM growth, PCM is composed of a single type of subunit that can either take an inactive form ($S_1$), or an enzyme-dependent active form ($S_1^*$), with $S_n$ representing

a centrosome with $n$ subunits. The single coarse-grained subunit ($S_1$) represents a composite of the scaffold-forming proteins (e.g. Spd-2 and Cnn in *Drosophila*), and the enzyme ($E$) represents the kinase (e.g. Polo in *Drosophila*). The inactive subunit can slowly bind and unbind from the PCM, while the enzyme-activated form can assemble faster (reactions 1 and 2 in *Figure 3B*). The subunit activation is carried out by the active form of the enzyme ($E^*$). Enzyme activation occurs in the PCM, and is thus centrosome size-dependent (reactions 3 and 4 in *Figure 3B*). A centrosome with a larger PCM thus produces active enzymes at a faster rate, and an increased amount of activated enzymes enhance centrosome growth. Thus, size-dependent enzyme activation generates a positive feedback in growth, which is shared between the centrosomes as the enzymes activated by each centrosome become part of the shared enzyme pool. This is in contrast to the autocatalytic growth model where the size-dependent positive feedback was exclusive to each centrosome.

A deterministic description for the growth of a single centrosome in a cell of volume $V_c$ is given by the coupled dynamics of centrosome size ($S_n$, number of incorporated subunits), the abundance of available active subunits ($S_1^*$) and the abundance of activated enzymes ($E^*$):

$$\frac{dS_n}{dt} = \frac{k^+}{V_c}S_1 + \frac{k^*}{V_c}S_1^* - k^- S_n \,, \tag{2}$$

$$\frac{dS_1^*}{dt} = \frac{k_1^*}{V_c}S_1 E^* - \frac{k^*}{V_c}S_1^* \,, \tag{3}$$

$$\frac{dE^*}{dt} = \frac{k_E^*}{V_c}S_n E - \frac{k_1^*}{V_c}S_1 E^* \,, \tag{4}$$

where $k^+$ and $k^*$ are the assembly rates for inactive and active form of the subunit, and $k^-$ is the disassembly rate. Here, $k^+$ represents the centriolar activity that can be different for the two centrosomes. The rates for PCM-dependent enzyme activation and enzyme-dependent subunit activation are given by $k_E^*$ and $k_1^*$ (*Figure 3B*). The condition for limiting component pool is imposed by substituting $S_1$ and $E$ with the constraints: $S_1 = N - S_n - S_1^*$, $E = N_E - E^* - S_1^*$, where $N$ and $N_E$ are the total amounts of subunits and enzymes, respectively.

## Model results and predictions

Using the above-described dynamics (*Equations 2–4* and *Figure 3B*), we performed stochastic simulations of a pair of centrosomes growing from a shared pool of enzymes and subunits. The resulting growth dynamics is sigmoidal, and lead to equally sized centrosomes (*Figure 3C*). Interestingly, the dynamics of the activated enzyme show an *activation pulse* at the onset of growth (*Figure 3C*). This pulse in the cytoplasmic concentration of active enzymes arises from the dynamics of enzyme activation by the PCM scaffold and its subsequent consumption by PCM subunits. The amplitude and the lifetime of the pulse depend on the difference in the timescales of enzyme activation and consumption (*Figure 3—figure supplement 1*). Notably, a pulse of centriolar Polo kinase density has been observed to initiate centrosome assembly in *Drosophila* (*Wong et al., 2022*). However, as we discuss later, further experiments are required to draw a direct correspondence between the centriolar Polo pulse and the pulse we observe here in the cytosolic active enzyme concentration. The experimentally observed Polo pulse is regulated by the abundance of the centriolar protein Ana1 (*Wong et al., 2022*), which controls the enzyme activation rate ($k_E^*$ in our model). Exploring the effect of the enzyme activation rate $k_E^*$, we observe increased pulse period and decreased pulse amplitude with decreasing enzyme activation rate (*Figure 3—figure supplement 1*). These results are similar to the experimentally observed effect of reduced Ana1, which reduces the overall rate of Polo activation in the centrosome (*Wong et al., 2022*).

Importantly, this model ensures robustness in centrosome size equality, with a negligible difference in steady-state size (~ 2% of mean size) that is independent of the initial size difference (*Figure 3D*). A linear stability analysis of the growth equations shows that the size difference between centrosomes decays exponentially, independent of the dynamics of subunit activation and enzyme activation (see Appendix 3 for details). The difference in steady-state size is a result of the fluctuations in the individual centrosome size dynamics, as evident from the distribution of the size difference (*Figure 3D* - inset). To further quantify the robustness in size regulation, we performed a statistical test by evaluating the Pearson correlation constant between the initial size difference and the final size difference

and find them to be uncorrelated (*Figure 3—figure supplement 2*). We find that the centrosome growth dynamics predicted by this model match really well with the experimental growth curves in *C. elegans* (*Decker et al., 2011*; *Figure 3E*).

Although centrosome growth in *C. elegans* is found to be sigmoidal, it has been suggested that centrosomes in *Drosophila* grow in a non-sigmoidal fashion (*Zwicker et al., 2014*). Although we could not find any direct quantitative measurement of centrosome size dynamics in *Drosophila* or other organisms, analysis of PCM assembly dynamics using fluorescence reporters show varying degrees of cooperativity during *Drosophila* development (*Wong et al., 2022*). We therefore sought to explore whether our catalytic growth model can also describe non-sigmoidal growth. To this end, we characterized the sigmoidal nature of the growth by fitting the dynamics of centrosome volume $V(t)$ to a Hill function of the form $At^\alpha/(B^\alpha + t^\alpha)$, where the coefficient $\alpha$ represents the strength of cooperativity. Our results show that the cooperative nature of growth depends on the interplay between the growth rate constant $k^+$ and the total enzyme concentration $[E]$, such that growth is sigmoidal ($\alpha \geq 2$) for larger $[E]$ and smaller $k^+$, and non-sigmoidal otherwise (*Figure 3F*).

While our model of shared catalysis considers a limiting pool of enzymes, a finite enzyme pool is not required for robust size control. To show this, we considered an unlimited pool of inactive enzymes ($E$), such that the cytoplasmic concentration of $E$ does not change over time (*Figure 3G*). The unlimited pool of inactive enzymes keeps producing activated enzymes via the centrosomes. The centrosome size reaches a steady-state when the subunit activation (via $E^*$) and subsequent growth is balanced by subunit disassembly from the centrosome (*Figure 3G* - inset). The size equality and cooperativity of growth remain intact in the presence of constant $[E]$ (*Figure 3H*). The prevalence of activated enzyme almost entirely depletes the inactive subunit pool and the centrosomes are in chemical equilibrium with the active subunit pool in the steady state (*Figure 3H* - inset).

Distinguishing between the autocatalytic and catalytic growth models from experimental data is not trivial as the qualitative features of growth and size scaling behaviors for a single centrosome are the same in both models. We find that the two models can be differentiated by measuring the correlation of the initial size difference with the final size difference of centrosome pairs. They are strongly correlated in the autocatalytic growth model with the sigmoidal growth curve but uncorrelated in the catalytic growth model (*Figure 3—figure supplement 2C-D*). The final size difference increases with decreasing the subunit pool size in catalytic growth model while no such relation was found in autocatalytic growth model (*Figure 3—figure supplement 2A-B*).

Finally, we extended our analysis beyond reaction-limited growth to examine how subunit diffusion affects catalytic centrosome growth, utilizing our spatially extended model. Our findings indicate that centrosome size equality, as predicted by the catalytic growth model, remains largely unaffected by variations in the diffusion constant or the separation distance between centrosomes (*Figure 3—figure supplement 3*).

## Cytoplasmic pool depletion regulates centrosome size scaling with cell size

Since our model for centrosome growth is limited by a finite amount of subunits, it is capable of capturing centrosome size scaling with cell size (*Figure 4A*), in excellent agreement with experimental data (*Decker et al., 2011*; *Zwicker et al., 2014*). However, the extent of organelle size scaling with cell size depends on the assembly rate and becomes negligible when the assembly rate is not significantly higher compared to the disassembly rate (*Figure 4B*). In particular, centrosome size scaling is connected to the extent of subunit pool depletion, such that the steady-state cytoplasmic fraction of the subunits is low when centrosome size scales with the cell size and higher otherwise (*Figure 4C*).

To understand how size scaling is regulated by the growth parameters, we derived a simplified analytical form (see Appendix 2) for the steady-state centrosome size given by

$$V = \frac{(E^*k_1^* + k^+)k^*\rho_0 V_c \delta v}{k^*(k^+ + k^- V_c) + E^*k_1^*(k^* + k^- V_c)}, \tag{5}$$

where $\delta v$ is the volume occupied by a centrosome subunit, $\rho_0$ is the total subunit density, and the enzymes are assumed to reach their steady-state abundance $E^*$ very fast. From the above expression, we can see that centrosome size $V$ will strongly scale with cell size $V_c$ when $k^+, k^* \gg k^- V_c$. This result is reflected in the phase diagram of size scaling (measured as the slope $\sim \mathrm{d}V/\mathrm{d}V_c$), which shows stronger

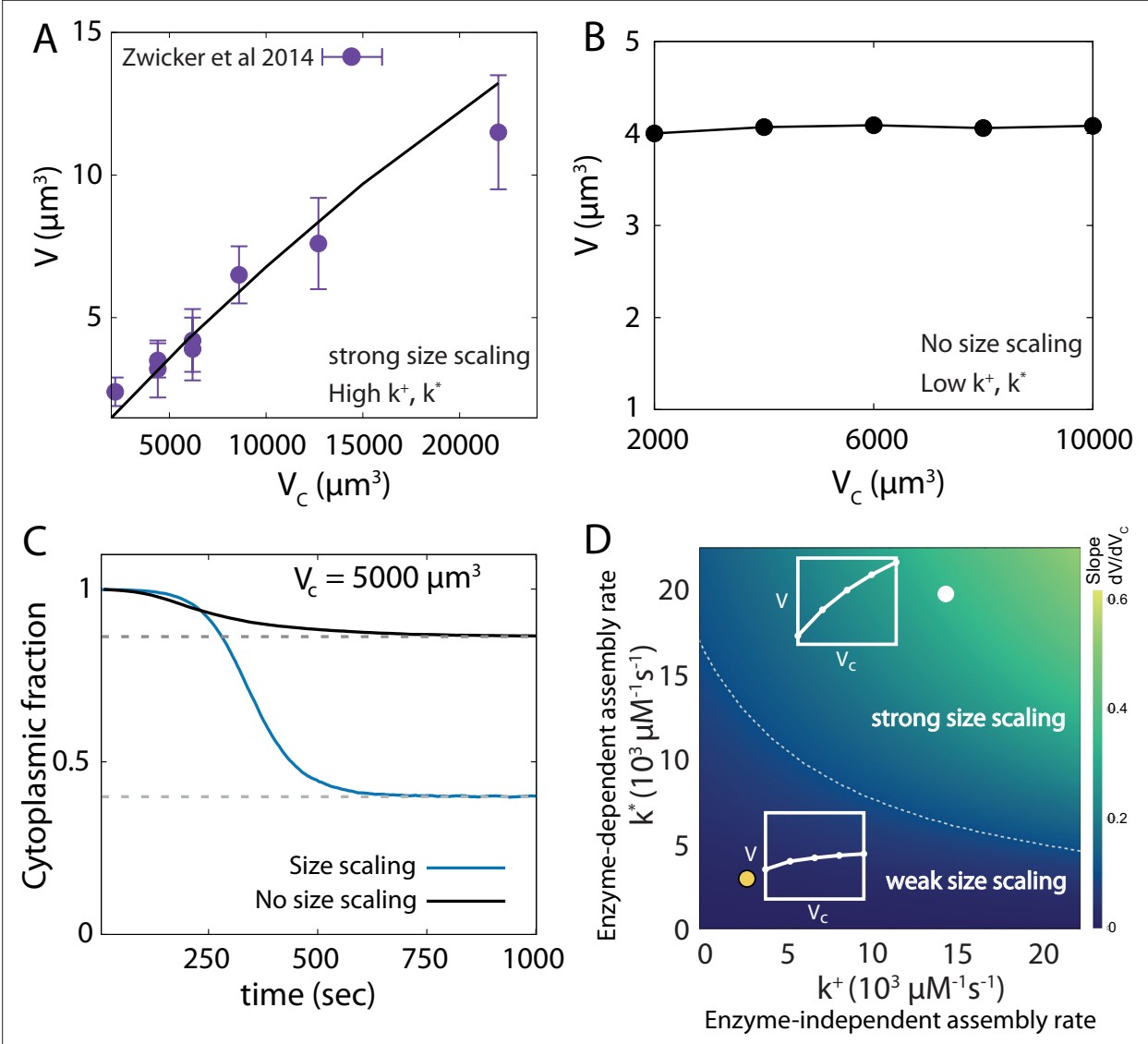

**Figure 4.** Centrosome size scaling with cell size. (**A**) Scaling of centrosome size with cell size obtained from the catalytic growth model (line) fitted to experimental data (points) in *C. elegans* embryo (*Zwicker et al., 2014*). (**B**) Centrosome size does not scale with cell size when the assembly rates are much lower compared to disassembly rate (i.e., $k^*, k^+ \lesssim k^- V_c$). (**C**) Dynamics of the cytoplasmic fraction of subunits ($S_1$ and $S_1^*$ combined) reveal significantly higher pool depletion in the size scaling regimes. The two curves correspond to the growth curves shown in panels A (blue) and B (black). The dashed lines are theoretical results obtained from the deterministic model. (**D**) An analytically obtained phase diagram of centrosome size scaling as functions of enzyme-dependent and enzyme-independent assembly rate constants. The color indicates the strength of size scaling (measured by $dV/dV_c$). The dashed gray line indicates the contour $dV/dV_c = 0.1$. Here the slope values are shown in $\delta v$ units. Insets: Characteristic size scaling behaviours. See *Table 1* for a list of parameter values. Parameters for panel B were obtained by tuning enzyme-dependent assembly rate and parameters for panel D were similar to panel A.

The online version of this article includes the following figure supplement(s) for figure 4:

**Figure supplement 1.** Centrosome size scaling and pool depletion in the catalytic growth model.

size scaling with increasing assembly rates (*Figure 4D*). The subunit pool depletion also increases with the assembly rates, reaching a state of almost complete depletion (i.e. $V \to \rho_0 V_c \delta v$) as we approach the regime of strong size scaling (see *Figure 4—figure supplement 1*).

It is important to note here that size scaling with cell size reported here is different from the linear size scaling predicted by the canonical limiting pool model (*Decker et al., 2011*; *Goehring and Hyman, 2012*). Robust size control for multiple centrosomes requires size-dependent negative feedback and with this feedback, the size scaling with cell size becomes a feature achieved in a range

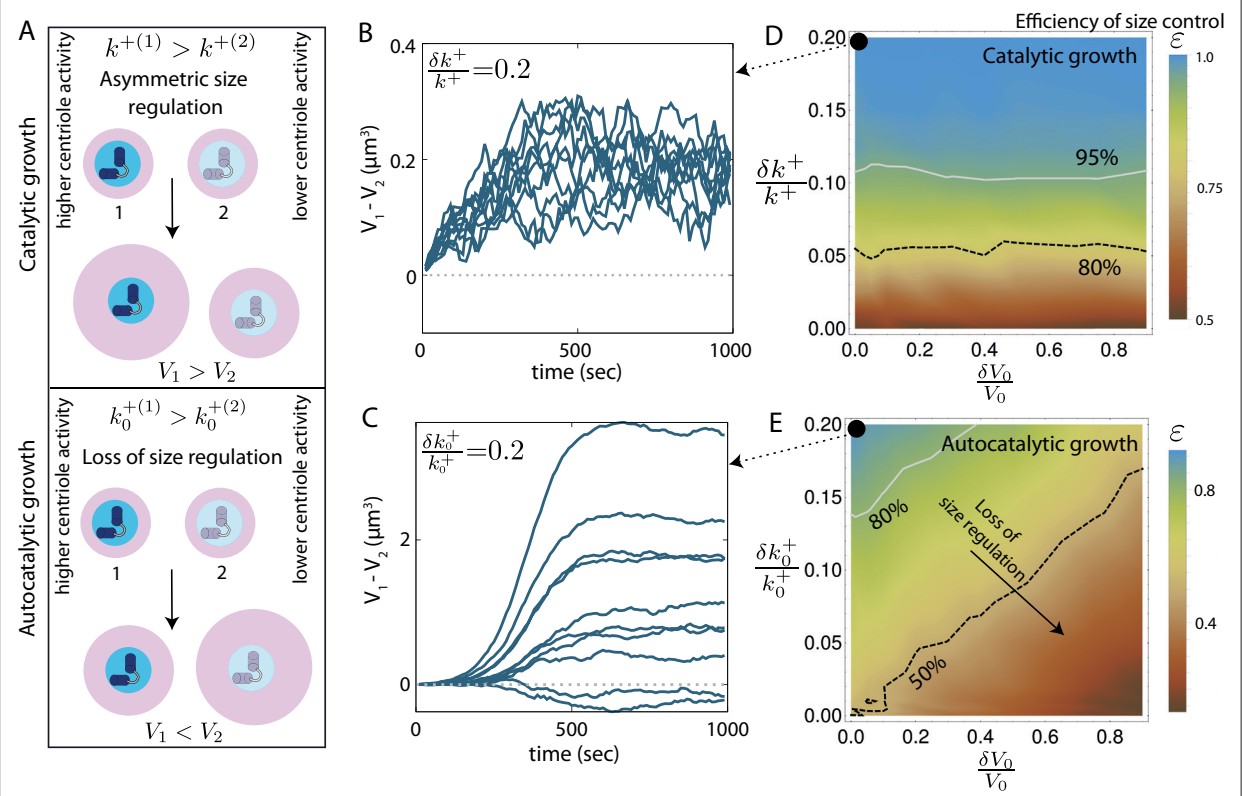

**Figure 5.** Control of centrosome size asymmetry via differential growth. (**A**) Schematic illustrating asymmetric size regulation via differential growth in the (top) catalytic growth model and (bottom) autocatalytic growth model. (**B,C**) Ten representative trajectories showing the dynamics of centrosome size difference ($V_1 - V_2$) for (**B**) catalytic growth model ($\delta k^+/k^+ = 0.2$), and (**C**) autocatalytic growth model ($\delta k_0^+/k_0^+ = 0.2$). The two centrosomes are initially of the same size. (**D**) Efficiency growth-rate-dependent control of centrosome size asymmetry ($\varepsilon = N^+/N_{\text{tot}}$) as a function of (normalized) initial size difference ($\delta V_0/V_0$) and (normalized) growth rate difference ($\delta k^+/k^+$), in the catalytic growth model. (**E**) Efficiency of growth-rate-dependent control of centrosome size asymmetry as a function of (normalized) initial size difference ($\delta V_0/V_0$) and (normalized) growth rate difference ($\delta k_0^+/k_0^+$), in the autocatalytic growth model. See **Table 1** for a list of model parameters. Parameter values for panels B and D were chosen to obtain typical steady-state centrosome size ($\sim 5\ \mu m^3$) and timescale of growth ($\sim 500\ S$).

of cell volumes by tuning growth rates. Interestingly, strong size scaling has been observed in *C. elegans* embryos (**Decker et al., 2011**), which are smaller in size ($\sim 10^4\ \mu m^3$) than *Drosophila* embryos ($\sim 10^6\ \mu m^3$) that do not exhibit size scaling with centrosome number (inferred from intensity data in **Wong et al., 2022**). This feature can be explained by our model in the regime of weaker size scaling, which is expected for larger system sizes (see Appendix 2 and **Figure 4—figure supplement 1**). Thus, the parameters of our model can be tuned to capture both sigmoidal and non-sigmoidal growth and strong or weak size scaling, without changing the nature of the molecular interactions that are largely conserved across organisms (**Raff, 2019**).

## Control of centrosome size asymmetry through differential growth

An essential aspect of centrosome size regulation is the modulation of centrosome size by centriole activity. In particular, it has been shown that the centrosome associated with a more active centriole will grow larger, resulting in centrosomes of unequal size (**Januschke et al., 2013**; **Conduit and Raff, 2010b**). Control of centriole activity-driven centrosome size asymmetry is important as this size asymmetry may play a crucial role in stem cell division as observed in *Drosophila* neuroblasts (**Conduit and Raff, 2010b**). We test the effectiveness of size regulation by studying the growth of a centrosome pair with different centriole activities, controlled by the values of the growth rate constants $k_0^+$ and $k^+$ for the autocatalytic (**Equation 1**) and the catalytic (**Figure 3B**) growth models, respectively (**Figure 5A**). For both the models, we bias the initial size of the centrosomes by assigning a smaller initial size ($V_0 - \delta V_0$) to the centrosome with a higher centriole activity (i.e., $k_0^{+(1)} = k_0^+ + \delta k_0^+$ or $k^{+(1)} = k^+ + \delta k^+$).

**Table 2.** Two-component growth model across organisms.

| Organism | Component $a$ | Component $b$ | Enzyme $E$ | Reference |
|---|---|---|---|---|
| Fly | DSpd-2/Spd-2 | Cnn | Polo | *Conduit et al., 2014b*; *Feng et al., 2017* |
| Worm | SPD-2 | SPD-5 | PLK-1 | *Woodruff et al., 2015*; *Wueseke et al., 2016* |
| *Xenopus*, Zebrafish and Mammals | Cep192 or Pericentrin | Cdk5Rap2/Cep215 | Plk1 | *Gomez-Ferreria et al., 2007*; *Fong et al., 2008*; *Lane and Nigg, 1996*; *Lee and Rhee, 2011*; *Doxsey et al., 1994*; *Aljiboury and Hehnly, 2023* |

We then simulate the growth of $N_{\text{tot}}$ centrosome pairs and quantify the efficiency ($\varepsilon$) of size control as the ratio of the number of cases ($N^+$) where the centrosome with higher growth rate ($k_0^+ + \delta k_0^+$ or $k^+ + \delta k^+$) becomes larger, to the total number of simulated pairs, $\varepsilon = N^+/N_{\text{tot}}$.

In the absence of any initial size difference ($\delta V_0 = 0$), the catalytic growth model shows better control of differential growth-induced size asymmetry (*Figure 5B*), while the autocatalytic growth model shows wide variations in centrosome size difference (*Figure 5C*). We find that the catalytic growth model ensures that the centrosome with a larger $k^+$ (higher centriole activity) end up being larger, irrespective of the initial size difference (*Figure 5D*). This illustrates robust control of centrosome size asymmetry by controlling differences in centriole activity. By contrast, in the autocatalytic growth model, the efficiency of size control monotonically decreases with increasing initial size difference, reflecting the lack of robustness in size control (*Figure 5E*).

## Multi-component centrosome model reveals the utility of shared catalysis on centrosome size control

One major postulate of the one-component PCM model was that the enzyme pool was shared between the two centrosomes rather than being localized to each. Here, we support this assumption using a more realistic multi-component centrosome model that allows us to model the specific interactions between the enzyme and the centrosome components, making it possible to study the relative dynamics of the two main scaffold formers. While we draw parallels between this model and the interactions observed in *Drosophila*, the model should be relevant to other organisms where similar pathways are in action via functionally similar proteins.

Based on recent studies (*Alvarez Rodrigo et al., 2019*; *Conduit et al., 2015b*), we model the centrosomes with two essential scaffold-forming proteins, $a$ and $b$, whose assembly into the PCM scaffold is regulated by the kinase $E$. The total size of the PCM scaffold, $S$, and the centrosome volume $V$ are given by $S = S(a) + S(b)$ and $V = V_a + V_b$, where $S(a)$ ($S(b)$) and $V_a$ ($V_b$) denote the contribution to the scaffold size (in number of subunits) and the centrosome volume by the component $a$ ($b$). The molecular identities of these key components are listed in *Table 2* for different organisms. In particular, for *Drosophila*, $a$ and $b$ can be identified as the scaffold forming proteins Spd-2 and Cnn, while $E$ represents the kinase Polo. It has been observed that Spd-2 and Cnn cooperatively form the PCM scaffold to recruit almost all other proteins involved in centrosome maturation (*Conduit et al., 2014b*). To effectively coordinate cooperative growth of the scaffold, Spd-2 proteins recruit the kinase Polo, which in turn phosphorylates Cnn at the centrosome (*Alvarez Rodrigo et al., 2019*). In the absence of Polo, Cnn proteins can bind to the scaffold but fall off rapidly, leading to diminished centrosome maturation (*Alvarez Rodrigo et al., 2019*; *Woodruff et al., 2015*).

We incorporated these experimental observations in our multi-component model as described in *Figure 6A*. We then test two different models for enzyme spatial distribution: (i) enzyme $E$ (Polo) is activated at each centrosome by the scaffold component $a$ (Spd-2), which then assembles the second component $b$ (Cnn) into the scaffold of that particular centrosome (for details see Appendix 5), and (ii) enzyme $E$ activated by the scaffold component $a$ is released in the cytoplasmic pool, promoting assembly of the $b$-scaffold at both centrosomes (for details see Appendix 5). In the first case, localized enzyme interaction exclusively enhances the growth of the individual centrosomes, creating an autocatalytic feedback that leads to size inequality of centrosomes (*Figure 6B*). Similar to model (*Equation 1*), the steady-state size difference between the two centrosomes increases with the increasing initial size difference, resulting in a failure of robust size control (*Figure 6—figure supplement 1*).

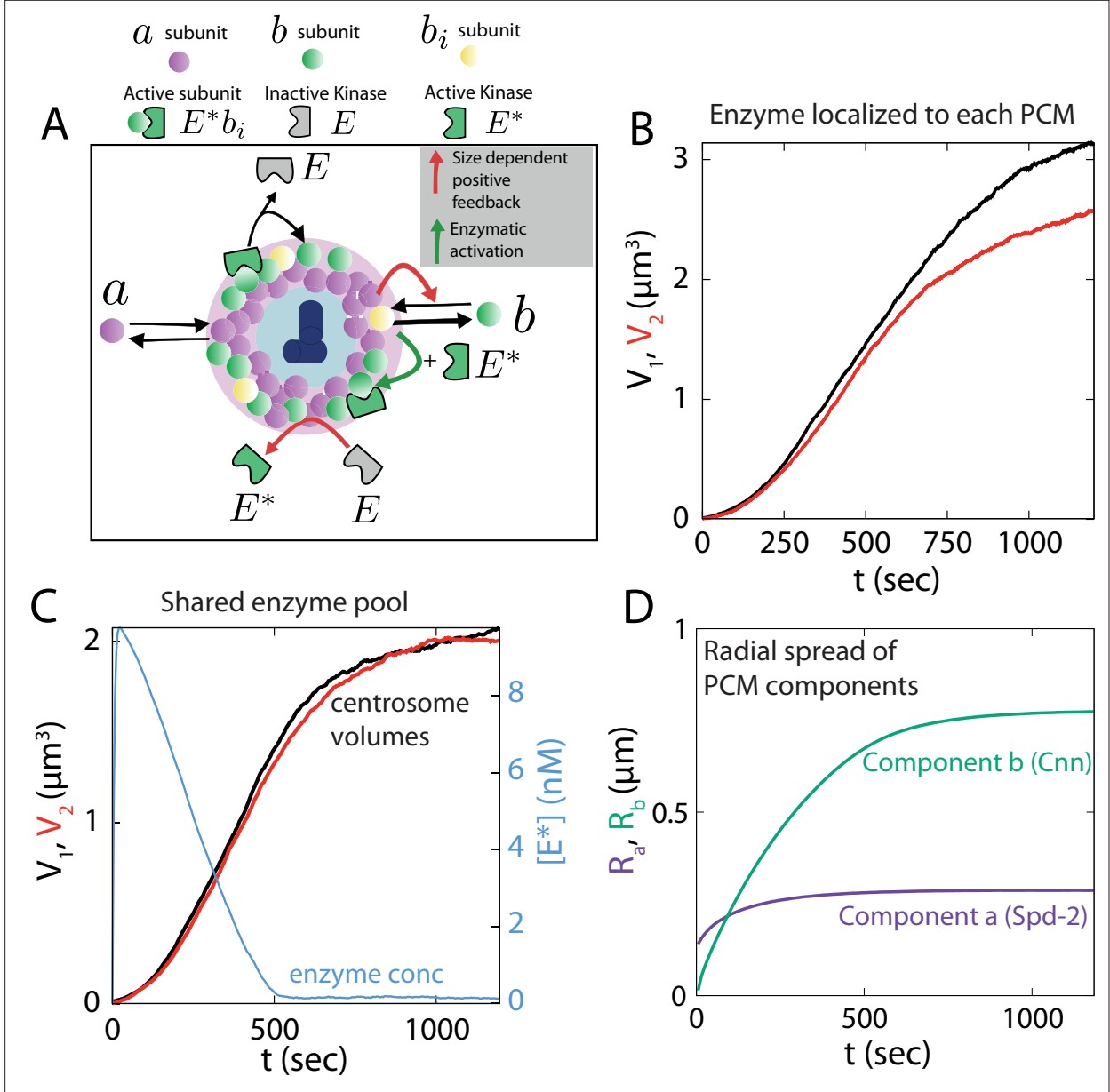

**Figure 6.** Multi-component model for centrosome growth. (**A**) Schematic of centrosome growth model driven by two scaffold components *a* and *b*, and enzyme E. *a* can bind the existing PCM independent of *b* or the enzyme $E$. The enzyme is activated by *a* in the scaffold, then released in the cytoplasm as $E^*$. The other scaffold former *b* binds to PCM in *a*-dependent manner in an intermediate form $b_i$ which can undergo rapid disassembly. The intermediate form $b_i$ can get incorporated in the *b*-scaffold by the active enzyme $E^*$ via forming an activated subunit form $E^* b_i$. The red arrows indicate the size-dependent positive feedback and the green arrow indicates the catalytic activity of the enzyme. (**B**) Centrosome size ($V_1, V_2$) dynamics for growth with localized enzyme. (**C**) Centrosome size ($V_1, V_2$) dynamics for growth with shared enzyme pool (black and red curve) and the pulse-like dynamics of activated enzyme concentration ($[E^*]$, blue curve). (**D**) Radial spread of the two scaffold former components *a* and *b* corresponding to the centrosome growth shown in panel-C. See *Table 1* for a list of parameter values. Parameter values for panel B & D were chosen to obtain typical steady-state centrosome size (~ 5 $\mu m^3$) and timescale of growth (~ 500 s).

The online version of this article includes the following figure supplement(s) for figure 6:

**Figure supplement 1.** Centrosome growth via localized enzyme activity.

**Figure supplement 2.** Centrosome growth via shared enzyme activity.

**Table 3.** Parameter values for two component growth via enzyme activity.

| | | | |
|---|---|---|---|
| $[\rho_b] = 0.25\,\mu M$ | $[\rho_b] = 0.5\,\mu M$ | $[\rho_b] = 0.01\,\mu M$ | $k_a^+ = 10\,\mu M^{-1}s^{-1}$ |
| $k_{b0}^+ = 0.5\,\mu M^{-1}s^{-1}$ | $k_{b0}^- = 0.01\,s^{-1}$ | $k_{aE}^+ = 5\times10^3\,\mu M^{-1}s^{-1}$ | $k_{Eb}^+ = 10^3\,\mu M^{-1}s^{-1}$ |
| $k_{b1}^+ = 10^4\,\mu M^{-1}s^{-1}$ | $k_{b1}^- = 5\times10^{-3}\,s^{-1}$ | $k_a^- = 5\times10^{-3}\,s^{-1}$ | |

We then considered the second case where the enzyme-mediated catalysis is shared between the growing centrosome pair. Experimental observations suggest a dynamic enzyme population around the centrosomes (*Mahen et al., 2011*; *Kishi et al., 2009*), with a turnover timescale much smaller than the scaffold forming proteins (*Conduit et al., 2014b*; *Conduit et al., 2015a*). These findings point towards the possibility that the enzyme is transiently localized in the centrosome during activation and the active enzyme is then released in the cytoplasmic pool that can enhance the growth of both the centrosomes (*Figure 6A*). We incorporate this shared catalysis mechanism in the second model where $a$ activates the enzyme to $E^*$ which then gets released in the cytoplasm, facilitating $b$-scaffold expansion in both the centrosomes (see Appendix 5 for details). This growth mechanism is able to robustly control centrosome size equality (*Figure 6—figure supplement 2*), giving rise to the characteristic sigmoidal growth dynamics (*Figure 6C*), where the first scaffold former $a$ is smaller in amount than the second, enzyme-aided component $b$. This difference in the abundances of $a$ and $b$ proteins, when translated into their respective radial spread from the centrosome center ($R \propto V^{1/3}$), bears close resemblance with the relative spread in Spd-2 and Cnn observed in the experiments, where the Cnn spread is twice as large as Spd-2 (*Conduit et al., 2014b*; *Alvarez Rodrigo et al., 2019*; *Figure 6D*). The active enzyme dynamics also resembles the observed pulse in Polo dynamics at the beginning of centrosome maturation (*Wong et al., 2022*; *Figure 6C*). Overall, the two-component model provides crucial insights into the role of shared catalytic growth on centrosome size control and lays the theoretical foundation for further investigations into the molecular processes that govern centrosome assembly.

## Discussion
### Autocatalytic feedback drives centrosome size inequality
In this article, we examined quantitative models for centrosome growth via assembly and disassembly of its constituent building blocks to understand how centrosome size is regulated during maturation. Although there is no generally accepted model for centrosome size regulation, previous studies *Conduit et al., 2014b*; *Zwicker et al., 2014*; *Conduit et al., 2014a*; *Alvarez Rodrigo et al., 2019*; *Woodruff et al., 2015*; *Conduit et al., 2015b* have suggested that centrosome assembly is cooperative and driven by a positive feedback mechanism. It has been quantitatively shown that an autocatalytic growth model (*Zwicker et al., 2014*) captures the cooperative growth dynamics of individual centrosomes as well as their size scaling features. However, as we showed here, autocatalytic growth does not guarantee the size equality of two centrosomes growing from a shared subunit pool. The resultant size inequality increases with the initial size difference between the centrosomes, indicating a lack of robustness in size control. This observation remains valid even within models where autocatalysis is not explicitly invoked, but emerges from positive feedback between PCM components (*Alvarez Rodrigo et al., 2019*). For instance, the positive feedback between Spd-2 and Cnn within *Drosophila* centrosomes results in the accumulation of more Cnn where Spd-2 is abundant. This, in turn, amplifies the retention of Spd-2 and binding of Cnn, culminating in a size-dependent positive feedback (akin to autocatalytic feedback) in PCM assembly. Given the current molecular understanding, it remains an open question whether localized assembly around the centriole, driven by autocatalytic feedback, is sufficient to furnish a robust mechanism for centrosome size regulation. It is important to note that the results shown in *Zwicker et al., 2014* indicate that the Ostwald ripening can be suppressed by the catalytic activity of the centriole, therefore stabilizing the centrosomes against coarsening by Ostwald ripening. However, if size discrepancy arises from the growth process (e.g. due to autocatalysis) the timescale of relaxation for such discrepancy is unclear from the above-mentioned result. We show that

for any appreciable amount of positive feedback, the system cannot achieve equal size in a physiologically relevant timescale (*Figure 2—figure supplement 3*).

## Model of centrosome pair growth via shared catalysis

Following recent experiments on the molecular mechanisms governing centrosome assembly, we constructed an enzyme-mediated catalytic growth model that not only describes cooperative growth behavior but also ensures robustness in size equality of the two maturing centrosomes. The enzyme Polo-like kinase (PLK1) that coordinates centrosome growth (*Conduit et al., 2014a*; *Woodruff et al., 2015*; *Ohta et al., 2021*; *Alvarez Rodrigo et al., 2019*), gets phosphorylated in the centrosome and has a much faster turnover rate than the centrosome scaffold forming proteins Spd-2 and Cnn (*Conduit et al., 2014b*). Experiments (~ 5 $\mu m^2$ $s^{-1}$ *Mahen et al., 2011*) and theoretical estimates (see Materials and methods) indicate high PLK1 diffusivity such that PLK1 transfer between the centrosome pair (assuming at a distance of ~ 5 – 10 $\mu m$) may occur within a few seconds which is much faster than the timescale of centrosome growth (~ 1000 s). This indicates that the kinase dynamics is not diffusion-limited, consistent with recent studies reporting negligible gradient in cytoplasmic Polo in *C. elegans* embryo (*Barbieri et al., 2022*). These insights led us to hypothesize that the kinase, once activated at the centrosome, could be released into the cytoplasm, becoming part of a shared pool of enzymes. This pool would then catalyze the growth of both centrosomes without any inherent bias. While we theoretically demonstrated that this mechanism of shared catalysis can robustly regulate centrosome size, it is important to acknowledge that the specific predictions concerning enzyme dynamics can only be validated through further experiments.

## Localized catalysis leads to centrosome size disparity

To further explore the role of enzymes in mediating centrosome growth and predict the consequence of an enzyme pool that is not shared equally by the two centrosomes, we extended our single-component model of catalytic growth to a multi-component model. This extended model incorporates the interactions PCM scaffold-forming proteins (Spd-2 and Cnn in *Drosophila*) and the enzyme Polo kinase. Using this model, we showed that localized catalysis by the enzyme—indicative of an unshared pool—leads to significcqnt size differences in the centrosomes. While direct experimental validation of a shared enzyme pool remains outstanding, it is intriguing to consider the findings that a centrosome-anchored Plk1 construct (Plk1-AKAP) induces anomalous centrosome maturation and defective spindle formation (*Kishi et al., 2009*).

## Enzyme-mediated size control

Our findings reveal that centrosome size increases with increasing enzyme concentration and that centrosome growth is inhibited in the absence of the enzyme (*Figure 6—figure supplement 2*). Since the activity of the Polo kinase is cell-cycle dependent (*Hamanaka et al., 1995*; *Uchiumi et al., 1997*), we further explored the dynamics of centrosome growth with a time-dependent dynamics of the enzyme. We found that centrosome growth can be triggered by switching on the enzyme dynamics and centrosome size was reduced when the enzyme was switched off (*Figure 6—figure supplement 2*). Importantly, it supported the experimental observation that a continuous Polo activity is required to maintain the PCM scaffold (*Mahen et al., 2011*; *Cabral et al., 2019*). Many key features of centrosome growth such as the sigmoidal growth curve and size scaling behavior can be modulated in our model by changing the growth rate constants and enzyme concentration, while conserving the underlying molecular mechanisms for assembly. This opens up the possibility that the catalytic growth model may be broadly relevant to other organisms where homologous proteins (*Table 2*) play similar functional roles in regulating centrosome growth (*Conduit et al., 2015b*).

## Testable model predictions

Aside from capturing the existing data on the dynamics of centrosome growth, our catalytic growth model makes specific predictions that can be tested in future experiments. Firstly, our model posits the sharing of the enzyme between the two centrosomes. This can potentially be experimentally tested through immunofluorescent staining of the kinase or by constructing FRET reporter of PLK1 activity (*Allen and Zhang, 2006*), where it can be studied if the active form of the PLK1 is found in the cytoplasm around the centrosomes indicating a shared pool of active enzyme. Another possible future

experiment can be performed based on photoactivated localization microscopy (PALM; *Sillibourne et al., 2011*) where fluorescently tagged enzyme can be selectively photoactivated in one centrosome and intensity can be measured at the other centrosome to find the extent of enzyme sharing between the centrosomes. It is important to to acknowledge that while we exclusively focused on Polo kinase as the sole enzyme, this shared catalytic activity might also involve other molecular players that interact with Polo, such as cyclin B/Cdk1 (*Kishi et al., 2009*). Moreover, our model provides explicit predictions regarding the enzyme's role in influencing centrosome size and growth. These predictions encompass the anticipated increase in centrosome size with increasing enzyme concentration, the ability to modify the shape of the sigmoidal growth curve, and the manipulation of centrosome size scaling patterns by perturbing growth rate constants or enzyme concentrations. Additionally, the model suggests inducing a shift from strong size scaling to weak size scaling through the reduction of PCM assembly rate or via cytoplasmic subunit pool depletion.

Secondly, an implication of our model is the robust regulation of centrosome size through catalytic PCM assembly during maturation. One direct avenue for testing this result is to observe the dynamics of two initially unequal-sized centrosomes during the early maturation phase. The catalytic growth model predicts that the final size difference of the centrosomes is uncorrelated to their initial size disparity while they are strongly correlated according to the autocatalytic growth model (*Figure 3—figure supplement 2*). The catalytic model also predicts the final size inequality will increase with decreasing subunit pool size. These predictions can be experimentally examined by inducing varying centrosome sizes at the early stage of maturation for different expression levels of the scaffold former proteins. It is important to note here that the initial size difference has to be induced while keeping the centrioles unaffected otherwise it may create size difference due to differences in centriole activity (*Januschke et al., 2013*; *Conduit and Raff, 2010b*; *Zwicker et al., 2014*). Experimentally validating these predictions will play a pivotal role in building a quantitative understanding of centrosome size regulation during mitosis and in clearly distinguishing the catalytic growth mechanism from the autocatalytic growth.

## Materials and methods

### Stochastic growth simulations

We use the *Gillespie, 1977* algorithm to simulate the stochastic growth of one or multiple structures from a common pool of subunits. At any time $t$ the Gillespie algorithm uses two random variables drawn from an uniform distribution ($r_1, r_2 \in \mathcal{U}(0, 1)$), and the instantaneous propensities for all of the possible reactions to update the system in time according to the defined growth law. The propensities of the relevant reactions, that is the assembly and disassembly rates of the $i^{th}$ structure are given by $K_i^{on}$ and $k_i^{off}$, respectively. For example, for the autocatalytic growth model described in *Equation 1*, these propensities are functions of subunit pool size ($N$) and structure size ($n_i$),

$$K_i^{\text{on}} = (k_0^+ + k_1^+ n_i) \left( \frac{N - \sum_{i=1}^{M} n_i}{V} \right) , \qquad (6)$$

$$K_i^{\text{off}} = k^- , \qquad (7)$$

where we are considering growth of $M$ structures from a shared pool. The Gillespie algorithm computes the time for the next reaction at $t + \tau$ given the current state of the system (i.e. the propensities for all reactions) at time $t$ where $\tau$ is given by-

$$\tau = \frac{1}{\sum_{i=1}^{C} \mathcal{R}_i} \log \left( \frac{1}{r_1} \right) , \qquad (8)$$

where $\mathcal{R}_i$ is the propensity of $i^{th}$ reaction and $C$ is the total number of all possible reactions. The second random variable $r_2$ is used to select the particular reaction ($j^{th}$ reaction) that will occur at $t + \tau$ time such that

$$\frac{\sum_{i=1}^{j-1} \mathcal{R}_i}{\sum_{i=1}^{C} \mathcal{R}_i} \leq r_2 < \frac{\sum_{i=1}^{j} \mathcal{R}_i}{\sum_{i=1}^{C} \mathcal{R}_i} . \qquad (9)$$

The condition for the first reaction ($j = 1$) is $0 \leq r_2 < \frac{\mathcal{R}_1}{\sum_{i=1}^{C} \mathcal{R}_i}$. The two steps defined by *Equation 8* and *Equation 9* are used recursively to compute the growth dynamics in time.

We used the Gillespie algorithm to find the stochastic trajectories of the above discussed deterministic (mass action kinetics) dynamics of the autocatalytic growth model and its various limits. See Catalytic growth in a shared enzyme pool ensures robust control of centrosome size for the corresponding chemical master equations. Similarly for the catalytic growth and two-component model, we find the stochastic trajectories via the Gillespie algorithm from the reactions given in *Figures 3B and 6A*.

## Subunit size estimation

Although we use single subunit and two subunit models of growth, we have used same value for the volume occupied by the subunit $\delta v$. We estimate the value of $\delta v$ from the molecular weight of SPD-5 which is 135 kDa (*Hamill et al., 2002*). Taking the protein mass density to be $1.4 \ \mathrm{gcc}^{-1}$(*Fischer et al., 2004*) and the PCM volume fraction to be ~ 0.1 (*Mahen et al., 2011*), we estimate the volume occupied by SPD-5 in PCM to be $0.1 \times 162 \times 10^{-7} \ \mu\mathrm{m}^3 \sim 2 \times 10^{-4} \ \mu\mathrm{m}$.

## Timescale of diffusion

We assume reaction-limited dynamics for centrosome maturation, meaning that the cytosolic diffusion of scaffold-forming proteins and the enzyme is much faster than their reaction rates. Here, we quantitatively discuss the timescales of protein diffusion and reaction based on their mass and fluorescent recovery after photobleaching (FRAP) data. The scaffold-forming proteins have a mass range of 100–150 kDa, while the enzyme mass is approximately 50–70 kDa. Using the Stokes-Einstein relation, which predicts that the diffusion constant scales inversely with protein radius ($R$), that is $D \sim R^{-1} \sim M^{-1/3}$ where $M$ is the protein mass, we estimate their diffusion constants. Based on the cytosolic diffusion constant of 30 $\mu m^2 s^{-1}$ for GFP (mass 30 kDa; *Milo et al., 2010*), we estimate diffusion constants of 17–20 $\mu m^2 s^{-1}$ for the scaffold-forming proteins and about 24 $\mu m^2 s^{-1}$ for the enzyme.

The separation distance between centrosomes ($d$) during maturation depends on the developmental stage of *C. elegans* and *Drosophila* embryos, but in later stages, it ranges between 5 and 10 $\mu m$ (*Decker et al., 2011*; *Alvarez Rodrigo et al., 2019*). Using the diffusion timescale $\tau_D = \frac{L^2}{6D}$, we estimate diffusion times of about 1 s for scaffold-forming proteins and 0.1–0.5 s for the enzyme. These diffusion times are significantly shorter than the turnover times observed in FRAP experiments, which are around 100 s for scaffold-forming proteins and 10 s for the enzyme in *Drosophila* (*Conduit et al., 2014b*; *Conduit and Raff, 2010b*) and *C. elegans* (*Woodruff et al., 2017*). This discrepancy suggests that diffusion of the relevant proteins and enzyme is considerably faster than their reaction rates, supporting the use of a reaction-limited model for studying the self-assembly of the PCM during centrosome maturation. Further experiments to directly measure diffusion constants of these proteins are necessary for a more detailed understanding of the role of diffusion in centrosome size regulation.

## Acknowledgements

We thank Jordan Raff and Zachary Wilmott for many useful discussions. SB acknowledges support from the National Institutes of Health (NIH R35 GM143042), National Science Foundation (NSF MCB-2203601) and the David Scaife Foundation.

## Additional information

### Funding

| Funder | Grant reference number | Author |
| --- | --- | --- |
| National Institutes of Health | NIH R35 GM143042 | Shiladitya Banerjee |
| David Scaife Foundation | | Shiladitya Banerjee |

| Funder | Grant reference number | Author |
|---|---|---|
| National Science Foundation | NSF MCB-2203601 | Shiladitya Banerjee |

The funders had no role in study design, data collection and interpretation, or the decision to submit the work for publication.

## Author contributions
Deb Sankar Banerjee, Conceptualization, Data curation, Formal analysis, Validation, Investigation, Visualization, Methodology, Writing – original draft, Writing – review and editing; Shiladitya Banerjee, Conceptualization, Supervision, Funding acquisition, Validation, Investigation, Visualization, Methodology, Writing – original draft, Project administration, Writing – review and editing

## Author ORCIDs
Deb Sankar Banerjee http://orcid.org/0000-0003-4452-7982
Shiladitya Banerjee https://orcid.org/0000-0001-8000-2556

Reviewer #1 (Public review): https://doi.org/10.7554/eLife.92203.3.sa1
Reviewer #2 (Public review): https://doi.org/10.7554/eLife.92203.3.sa2
Author response https://doi.org/10.7554/eLife.92203.3.sa3

# Additional files

## Supplementary files
MDAR checklist

## Data availability
The current manuscript is a computational study, so no data have been generated for this manuscript. Modelling code is available on Github (copy archived at *Banerjee, 2024*).

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

## Appendix 1

### Autocatalytic growth model

Here we present the derivation of the autocatalytic growth model of centrosomes developed by *Zwicker et al., 2014*, where centrosomes are described as phase-segregated liquid droplets. We develop an equivalent kinetic model and study the centrosome size evolution using the corresponding stochastic description. Specifically, we consider PCM droplet growth in the limit of strong phase segregation and reaction-limited growth (i.e., fast diffusion of PCM components). The growth of the centrosome volume $V$ is given by

$$\frac{dV}{dt} = (k\phi_1^A - k_{BA})V + Q\frac{\phi_1^A}{\psi_-} + k_{AB}\frac{\phi_0^A V_c}{n_c \psi_-} \tag{10}$$

where $\phi^A$ and $\phi^B$ are the volume fractions for the soluble ($A$) and the phase segregated ($B$) forms of the PCM components, respectively, and $\psi_-$ is the volume fraction of $B$ inside the droplet. Here $k_{AB}$, $k_{BA}$ and $k$ are the reactions rates for $A \rightarrow B$, $B \rightarrow A$ and $AB \rightarrow B$ respectively. The bulk volume fraction of the $A$ and $B$ forms, away from the droplet, is given by $\phi_0^A$ and $\phi_0^B$ respectively and $\phi_1^A$ is the volume fraction of $A$ inside the droplet. The chemical activity of the centriole, centrosome number, and volume of the cell are given by $Q$, $n_c$, and $V_c$, respectively.

To model the growth of a pair of centrosomes, we write the equation for size kinetics in terms of the number of incorporated subunits. As assumed by *Zwicker et al., 2014*, we neglect spontaneous production of phase segregated form $B$ away from the centriole ($k_{AB} = 0$), and the volume fraction of $B$ inside the droplet is considered to be unchanged, i.e., $\psi_-$ = constant. The volume fractions of $A$ in the bulk and inside the droplet are given by

$$\phi_0^A = \bar{\phi} - \psi_- \left(\frac{V_1 + V_2}{V_c}\right) \tag{11}$$

$$\phi_1^A = (1 - \psi_-)\phi_0^A \tag{12}$$

where $\bar{\phi}$ is the average volume fraction of the total PCM material and $V_{1,2}$ are size of the two centrosomes/droplets. The average volume fraction is given by $\bar{\phi} = \frac{V_A + V_B}{V_c} = \frac{N\delta v}{V_c}$, where $N$ is the total number of PCM subunits that remains constant during the droplet growth and $\delta v$ is the volume occupied by a single subunit. The volume fraction of subunits inside the droplet is given by $\psi_- = \frac{n_B^i \delta v}{V_i}$, where $n_B^i$ is the number of $B$ subunits inside the $i^{th}$ droplet (centrosome) and $V_i$ is the volume of that centrosome. We can now rewrite $\phi_0^A = \frac{N_{av}\delta v}{V_c}$ and $\phi_1^A = (1 - \psi_-)\left(\frac{N_{av}\delta v}{V_c}\right)$ where $N_{av} = N - n_B^1 - n_B^2$ is the available amount of subunits that can contribute to the droplet growth. Finally using the definition of $\psi_-$ we get the $n_B^i$ dynamics given by

$$\left(\frac{\delta v}{\psi_-}\right)\frac{dn_B^i}{dt} = \frac{dV_i}{dt}. \tag{13}$$

We shall drop the index $B$ and write $n_B^i \rightarrow n_i$ and rewrite the above equation as

$$\frac{dn_i}{dt} = (1 - \psi_-)\left(k\delta v\, n_i + Q\right)\left(\frac{N_{av}}{V_c}\right) - k_{BA}n_i. \tag{14}$$

This above description then can be rewritten as

$$\dot{n}_i = (k_0^+ + k_1^+ n_i)\left(\frac{N_{av}}{V_c}\right) - k^- n_i \tag{15}$$

where $k_0^+ = (1 - \psi_-)Q$, $k_1^+ = (1 - \psi_-)k\delta v$ and $k^- = k_{BA}$. This final equation (*Equation 15*) can be recognized as the main text *Equation 1*.

The stochastic growth corresponding to the above discussed deterministic growth dynamics can be described by a chemical master equation for the joint probability $P(n_1, n_2, t)$:

$$\frac{dP(n_1, n_2, t)}{dt} = (k_0^+ + k_1^+(n_1 - 1))\left(\frac{N - (n_1 + n_2 - 1)}{V_c}\right)P(n_1 - 1, n_2, t)$$

$$+ (k_0^+ + k_1^+(n_2 - 1))\left(\frac{N - (n_1 + n\_2 - 1)}{V_c}\right)P(n_1, n_2 + 1, t)$$

$$+ k^-(n_1 + 1)P(n_1 + 1, n_2, t) + k^-(n_2 + 1)P(n_1, n_2 + 1, t)$$

$$- \left((k_0^+ + k_1^+ n_1)\left(\frac{N - (n_1 + n_2)}{V_c}\right) + (k_0^+ + k_1^+ n_2)\left(\frac{N - (n_1 + n_2)}{V_c}\right) + k^- n_1 + k^- n_2\right)$$

$$P(n_1, n_2, t)$$

(16)

where $P(n_1, n_2, t)$ is the probability of having centrosomes with size $n_1 \, \delta v$ and $n_2 \, \delta v$ at time $t$. We numerically simulate the growth dynamics using *Gillespie, 1977* first algorithm , with the transition probabilities described in the master equation (*Equation 17*).

Studying the phase portrait of the centrosome pair dynamics reveals the origin of size inequality (*Figure 2—figure supplement 3A-D*). In the regime of low $k_0^+$ values, the growth is strongly autocatalytic and the phase portrait shows a quasi line-attractor where the solutions which are away from the $V_1 = V_2$ line get trapped and cannot reach the fixed point in a biologically relevant timescale (*Figure 2—figure supplement 3A, C*). With increasing $k_0^+$ value (weakly cooperative or non-cooperative growth) the quasi line-attractor goes away and solutions reach the equal size fixed point quickly (i.e., size inequality is small) but the sigmoidal nature is absent in the size dynamics (*Figure 2—figure supplement 3E, F*) in this regime.

## Limiting cases of the autocatalytic growth model

Here we consider the two limits of the autocatalytic growth model (*Equation 15*): purely autocatalytic limit ($k_0^+ = 0$) and purely non-autocatalytic limit ($k_1^+ = 0$). First, we shall consider the purely autocatalytic limit where the resulting growth dynamics can be written as

$$\dot{n_1} = k^+ n_1\left(\frac{N_{av}}{V_c}\right) - k^- n_1$$

$$\dot{n_2} = k^+ n_2\left(\frac{N_{av}}{V_c}\right) - k^- n_2$$

(17)

where $k^+ = k_1^+$ is the assembly rate constant. The concentration of available subunits is given by $\rho = \left(N - \sum_{i=1}^2 n_i\right)/V_c = \frac{N_{av}}{V_c}$. This resulting growth model can be described as growth via the assembly and disassembly of subunits throughout the volume. The steady-state solution to the above equations (i.e., $\dot{n_1^*} = 0$ and $\dot{n_2^*} = 0$) constitute a system of underdetermined equations that give rise to a line attractor (a line of fixed points) given by $n_1^* + n_2^* = N - \frac{k^- V_c}{k^+}$.

The stochastic description of the growth can be given by the following chemical master equation for the joint probability

$$\frac{dP(n_1, n_2, t)}{dt} = k^+(N - (n_1 + n_2 - 1))P(n_1 - 1, n_2, t) + k^+(N - (n_1 + n_2 - 1))P(n_1, n_2 - 1, t)$$

$$+ k^-(n_1 + 1)P(n_1 + 1, n_2, t) + k^-(n_2 + 1)P(n_1, n_2 + 1, t)$$

$$- \left(k^+(N - (n_1 + n_2)) + k^+(N - (n_1 + n_2)) + k^- n_1 + k^- n_2\right)P(n_1, n_2, t).$$

where $P(n_1, n_2, t)$ is the probability of having centrosomes with sizes $n_1 \, \delta v$ and $n_2 \, \delta v$ at time $t$. In this purely autocatalytic limit, the size of a single centrosome is well regulated but two centrosomes growing from a shared subunit pool exhibit large size inequality (*Figure 2—figure supplement 1A*). The phase portrait for a growing pair of centrosomes reveals the line-attractor where the solutions get trapped away from the equal size point (*Figure 2—figure supplement 1B*). This leads to a lack of robustness in size regulation as size inequality (|$\delta V$|) increases with increasing initial size difference $\delta V_0$ (*Figure 2—figure supplement 1C*).

The purely non-cooperative growth dynamics can be described by

$$\dot{n_1} = k^+ n_1 \left( \frac{N_{av}}{V_c} \right) - k^- n_1$$

$$\dot{n_2} = k^+ n_2 \left( \frac{N_{av}}{V_c} \right) - k^- n_2$$

(18)

where $k^+ = k_0^+$ is the assembly rate constant. We can calculate a unique stable fixed point of equal size ($V_1 = V_2 = n^* \delta v$), given by $n^* = \frac{k^+ N}{k^- V_c + 2k^+}$. This model describes centrosome growth via localized assembly (i.e., centrosome size-independent assembly) and distributed disassembly throughout the volume. The stochastic description of the growth can be given by the following chemical master equation for the joint probability

$$\frac{dP(n_1, n_2, t)}{dt} = k^+ (N - (n_1 + n_2 - 1)) P(n_1 - 1, n_2, t) + k^+ (N - (n_1 + n_2 - 1)) P(n_1, n_2 - 1, t)$$

$$+ k^- (n_1 + 1) P(n_1 + 1, n_2, t) + k^- (n_2 + 1) P(n_1, n_2 + 1, t)$$

$$- \left( k^+ (N - (n_1 + n_2)) + k^+ (N - (n_1 + n_2)) + k^- n_1 + k^- n_2 \right) P(n_1, n_2, t).$$

This growth model exhibits robust size control for a centrosome pair, giving rise to centrosomes of similar size even in the presence of large initial size differences (*Figure 2—figure supplement 2A, B*). Unlike the case of purely autocatalytic growth, there is no line-attractor in this model and the solutions leads to the fixed point with equal-sized centrosomes (*Figure 2—figure supplement 2C*). Though this model leads to robust size control, there is no cooperativity in growth and the resulting size dynamics is non-sigmoidal. Steady-state probability distribution of size can be analytically obtained (as a finite sum) for these two growth models given by *Equation 18* and *Equation 19* and shows that the size distribution is approximately uniform in the case of purely autocatalytic growth and narrowly distributed in the case of purely non-autocatalytic growth *Banerjee and Banerjee, 2022*.

## Appendix 2

### Deterministic description of catalytic growth model and centrosome size scaling predictions

Here we present the deterministic description of the catalytic growth model presented in the main text. We begin by describing the dynamics of a single centrosome of $n$ subunits, whose size is denoted as $S_n$. The subunits in the cytoplasmic pool can be either in the active form $S_1^*$ or the inactive form $S_1$. The enzymes can also be in two forms - an active enzyme pool of abundance $E^*$, and an inactive enzyme pool of size $E$, respectively. We can completely describe centrosome growth from the dynamics of $S_n$, $S_1^*$ and $E^*$ by substituting $S_1$ and $E$ with the constraints arising from limited pools: $S_1 = N - S_n - S_1^* = N_{av}$ and $E = N_E - E^* - S_1^* = N_{av}^E$. Here $N = \rho_0 V_c$ is the total amount of subunits, $N_E = \rho_E V_c$ is the total amount of enzymes and $V_c$ is the volume of the cell. The dynamic rate equations are given by

$$\dot{S}_n = \frac{k^+}{V_c}N_{av} + \frac{k^*}{V_c}S_1^* - k^-S_n,$$

$$\dot{S}_1^* = \frac{k_1^*}{V_c}N_{av}E^* - \frac{k^*}{V_c}S_1^*, \qquad (19)$$

$$\dot{E}^* = \frac{k_E^*}{V_c}S_nN_{av}^E - \frac{k_1^*}{V_c}N_{av}E^*.$$

The analytical steady-state solutions of the above equations are cumbersome and not very insightful. Assuming that all centrosomes attain equal sizes (using results presented in the main text), then the above equations can be easily extended to describe the growth of $M$ centrosomes, given by

$$\dot{S}_n = \frac{k^+}{V_c}N_{av} + \frac{k^*}{V_c}S_1^* - k^-S_n,$$

$$\dot{S}_1^* = \frac{k_1^*}{V_c}N_{av}E^* - \frac{k^*}{V_c}MS_1^*, \qquad (20)$$

$$\dot{E}^* = \frac{k_E^*}{V_c}MS_nN_{av}^E - \frac{k_1^*}{V_c}N_{av}E^*,$$

where $N_{av} = N - MS_n - S_1^*$.

To understand how centrosome size scales with cell size and centrosome number, we make a simplifying assumption of fast enzyme activation dynamics, i.e., the active enzyme concentration reaches steady state very fast. This will enable us to obtain useful analytical solutions for steady state centrosome size. Solving for $S_n$ and $S_1^*$, assuming a steady state enzyme abundance $E^*$, we obtain the steady-state centrosome size

$$V = \frac{(E^*k_1^* + k^+M)k^*\rho_0V_c\delta v}{k^*M(k^+M + k^-V_c) + E^*k_1^*(k^*M + k^-V_c)}. \qquad (21)$$

With $M = 1$, we derive the expression shown in the main text. Centrosome size scaling with cell size can be obtained when $k^+M \gg k^-V_c$ or $k^*M \gg k^-V_c$. To obtain a quantitative measure of size scaling, we evaluate the slope of centrosome size with cell size given by

$$\frac{dV}{dV_c} = \frac{(E^*k_1^* + k^+M)^2k^{*2}M\rho_0\delta v}{\left(k^*M(k^+M + k^-V_c) + E^*k_1^*(k^*M + k^-V_c)\right)^2}. \qquad (22)$$

These results show that the extent of size scaling will become weaker for larger system size (or cell/organism size) if other growth rates are similar (*Figure 4—figure supplement 1A*). We can estimate the extent of pool depletion using the cytoplasmic fraction of subunits at steady-state (combined amount of $S_1$ and $S_1^*$), given by

$$f_c = \frac{N - M(V/\delta v)}{N}$$

$$= \frac{(E^* k_1^* + k^* M) k^- V_c}{k^* M(k^+ M + k^- V_c) + E^* k_1^* (k^* M + k^- V_c)} .$$

(23)

The subunit pool depletion is directly connected to the extent of size scaling with stronger size scaling occurring at higher pool depletion (**Figure 4—figure supplement 1B**).

## Relation between pool depletion and size scaling

Here consider a simple example of the growth of $M$ structures in a shared pool of $N$ subunits in volume $V_c$. We assume a linear size-dependent negative feedback to growth rate. Using the prior knowledge of robust size control in this case, we can write down the size ($n$ in subunits) dynamics in terms of a single structure:

$$\dot{n} = \frac{k^+(N - Mn)}{V_c} - k^- n .$$

(24)

Notice that the second term in the RHS $\frac{k^+ Mn}{V_c}$ embodies the pool depletion rate. We can define a bare rate of pool depletion as $\mathcal{D} = k^+ M/V_c$. The steady-state size is given by:

$$n = \frac{k^+ \rho_0 V_c}{k^+ M + k^- V_c}$$

$$= \frac{k^+ \rho_0}{\frac{k^+ M}{V_c} + k^-} ,$$

(25)

where $N = \rho_0 V_c$. It thus becomes clear that we obtain strong size scaling with system size and structure number in the regime $\mathcal{D} = \frac{k^+ M}{V_c} \gg k^-$, when the depletion rate is much higher that the disassembly rate ($k^-$) or pool replenishing rate. We can identify the two pool depletion rates in catalytic growth from $S_n$ and $S_1^*$ dynamics in **Equation 20** as $\mathcal{D}_1 = \frac{k^+ M}{V_c}$ and $\mathcal{D}_2 = \frac{k^* M}{V_c}$. Thus, the condition for strong size scaling comes from the condition of strong pool depletion $\frac{k^+ M}{V_c}, \frac{k^* M}{V_c} \gg k^-$, similar to the example case described above.

Our analysis indicates that the size scaling of intracellular organelles is a result of fine-tuning of growth parameters rather than due to the physical constraint of having a limited pool of building blocks. Structures growing in a shared limited pool of subunits with size-dependent negative feedback (which is the case in centrosome growth) require information of the system size ($V_c$) and number of structures ($M$) to scale with these quantities. This information is encoded in the depletion rate, i.e., $\mathcal{D} = k^+ M/V_c$. Hence, when strong depletion of the subunit pool sets the structure size, it enables the sensing of the system size and structure number, resulting in strong size scaling.

## Centrosome size scaling in *Drosophila* and *C. elegans*

During the embryonic development of *C elegans*, as the centrosome number increases with the progression of the development, centrosome size decreases as $\sim 1/M$, where $M$ is the centrosome number **Decker et al., 2011**. Interestingly, during the development of *D Melanogaster* centrosome size scaling with centrosome number is negligible in the cycles 11 to 12 (**Wong et al., 2022**) during which centrosome number increases by $\sim 1000$. This apparent disparity in behaviour of centrosome growth can be understood from the difference in size between the two embryos. The *Drosophila* embryo is much larger in size at $500\,\mu m \times 180\,\mu m$ (L x W) compared to the C *elegans* embryo which is $50\,\mu m \times 30\mu m$ (L x W). Our theory predicts strong centrosome size scaling in the initial cycles of C *elegans* embryo and almost no size scaling for *Drosophila* embryo during cycles 11 to 12 (**Figure 4—figure supplement 1C-D**). Thus, explaining how distinctly different quantitative features of growth can emerge from the same underlying mechanisms.

## Appendix 3

### Linear stability analysis of the growth models

Condition for size inequality in autocatalytic growth

We consider two centrosomes growing according to the autocatalytic growth model described in the main text and in *Equation 15*,

$$\dot{n}_1 = (k_0^+ + k_1^+ n_1)\left(\frac{N - n_1 - n_2}{V_c}\right) - k^- n_1$$

$$\dot{n}_2 = (k_0^+ + k_1^+ n_2)\left(\frac{N - n_1 - n_2}{V_c}\right) - k^- n_2 \,. \tag{26}$$

We can derive the equations governing the difference of size $\Delta = n_2 - n_1$ and the sum of the two centrosome sizes $S = n_1 + n_2$ from the above equations:

$$\dot{\Delta} = \frac{k_1^+}{V_c}(N - S)\Delta - K^- \Delta$$

$$\dot{S} = (2k_0^+ + k_1^+ S)\frac{N - S}{V_c} - k^- S \,. \tag{27}$$

As we are interested in the dynamical behavior of size difference that arises from autocatalytic growth, we linearized the equation for $S$ at small times when $S$ is small, allowing us to approximate $2k_0^+ + k_1^+ S \sim 2k_0^+$. Next, we can solve for $S(t)$ as

$$S(t) = 2k_0^+ \tau_S (N/V_c)(1 - e^{-\frac{t}{\tau_S}}) \tag{28}$$

where $\tau_S = \frac{V_c}{2k_0^+ + k^- V_c}$ is the timescale for growth. Plugging this solution for $S$ in the $\Delta$ equation (*Equation 27*), we can solve for $\Delta$ as

$$\Delta(t) = \Delta(0)e^{\beta} \tag{29}$$

where

$$\beta = \frac{(k_1^+ N - (2k_0^+ + k^- V_c))k^- t}{2k_0^+ + k^- V_c} - \frac{2k_0^+ k_1^+ N}{(2k_0^+ + k^- V_c)^2}(1 - e^{-\frac{t}{\tau_S}}) \,. \tag{30}$$

As the contribution of the second term exponentially decreases over time, we focus on the first term on the right-hand side. The dynamics of the size difference is primarily determined by the sign of the term $\lambda = k_1^+ N - (2k_0^+ + k^- V_c)$. Such that, for $\lambda > 0$ the size difference will increase over time while it will decrease if $\lambda < 0$. Thus, this linear analysis predicts the condition for size inequality in autocatalytic growth to be $k_1^+ N > (2k_0^+ + k^- V_c)$.

Catalytic growth shows monotonic decay of size inequality

Here we present a linear stability analysis of the catalytic growth model given in *Equation 19*. We consider two centrosomes, $S_n^{(1)}$ and $S_n^{(2)}$, growing in a shared pool of subunits. The active and inactive forms are denoted by $S_1^*$ and $S_1$, respectively, and the active and inactive enzyme pools are denoted by $E^*$ and $E$, respectively. We consider small perturbations of the concentration values around their respective steady-state values (represented by a superscript 0), e.g., $S_1 \to S_1^0 + \delta S_1$, $S_1^* \to S_1^{*0} + \delta S_1^*$, etc. To linear order in the perturbations, the dynamics of the system are given by:

$$\delta\dot{S}_n^{(1)} = \frac{k^*}{V_c}\delta S_1^* - k^{-\delta}S_n^{(1)} - \frac{k^+}{V_c}(\delta S_n^{(1)} + \delta S_n^{(2)} + \delta S_1^*),$$

$$\delta\dot{S}_n^{(2)} = \frac{k^*}{V_c}\delta S_1^* - k^{-\delta}S_n^{(2)} - \frac{k^+}{V_c}(\delta S_n^{(1)} + \delta S_n^{(2)} + \delta S_1^*),$$

$$\delta\dot{S}_1^* = \frac{k_1^*}{V_c}(S^0\delta E^* - E^{*0}(\delta S_n^{(1)} + \delta S_n^{(2)} + \delta S_n^*)) - \frac{k^*}{V_c}\delta S_1^*, \tag{31}$$

$$\delta\dot{E}^* = \frac{k_E^*}{V_c}(E^0(\delta S_n^{(1)} + \delta S_n^{(2)} - (S_n^{(1)0} + S_n^{(2)0})\delta E^*))$$

$$- \frac{k_1^*}{V_c}(S_1^0\delta E^* - E^{*0}(\delta S_n^{(1)} + \delta S_n^{(2)} + \delta S_1^*)).$$

We can express the perturbation to centrosome size difference as $\delta\Delta = \delta S_n^{(2)} - \delta S_n^{(1)}$. Using *Equation 31*, we arrive at

$$\delta\dot{\Delta} = -k^-\delta\Delta \tag{32}$$

which shows an exponential decay of the size difference, dependent on a single timescale $1/k^-$ determined by the disassembly rate of the subunits from the centrosome. This result reflects the robust regulation of size equality in the catalytic growth model. Note, that we could also use the full set of equations (*Equation 19*) to arrive at a similar equation for $\Delta = S_n^{(2)} - S_n^{(1)}$ with the same exponential form for the decay of $\Delta$.

## Appendix 4

### Effect of subunit diffusion on centrosome size regulation

To explore the effect of diffusion on centrosome size regulation, we developed a spatially extended model of centrosome growth. We relaxed the assumption of reaction-limited growth and explicitly modeled the reactions of subunits within a 3D volume, represented as a collection of small voxels. Using a simple approach (*Bernstein, 2005*; *Erban et al., 2007*), we treated diffusion as a reaction process that enables monomer transport between voxels, with the reaction rate given by $k_D = \frac{D}{\delta x^2}$, where $D$ is the diffusion constant and $\delta x$ is the voxel size. This diffusion process was incorporated into our stochastic growth models based on the Gillespie algorithm. Although centrosomes move apart during the G2/M phase of the cell cycle, we ignored their motion for simplicity, noting that centrosome movement is not likely diffusive and would require careful modeling. In this model, centrosomes were placed at distinct positions, and we examined their growth under different diffusion constants and inter-centrosomal distances (*Figure 2—figure supplement 4A*). For simplicity, we assumed that all subunits, whether active or inactive, and the enzymes shared the same diffusion constant in the catalytic growth model. While this method is exact, it is computationally expensive, so we reduced computational costs by using a smaller pool size and a smaller system size of approximately 9–10 μm.

The qualitative results of the centrosome size regulation in the autocatalytic growth model remain consistent in the presence of explicit subunit diffusion. The ensemble average of the final size difference decreases with increasing diffusive timescales, meaning lower diffusion constants or greater distances between centrosomes lead to smaller size inequalities (*Figure 2—figure supplement 4C-D*). At low diffusion constants and large separation distances between centrosomes, the size inequality is small (*Figure 2—figure supplement 4B-D*), though the characteristic sigmoidal shape of the growth curve is lost in this regime (*Figure 2—figure supplement 4B*).

Incorporating explicit subunit diffusion ($S_1$, $S_1^*$, $E$, and $E^*$) in the catalytic growth model does not significantly alter the growth characteristics (*Figure 3—figure supplement 2*). Centrosome growth retains its sigmoidal behavior, and the final size difference remains small across varying diffusion constants and centrosome separations (*Figure 3—figure supplement 2A, B*). Additionally, a large initial size difference did not result in substantial changes in the final size difference (*Figure 3—figure supplement 2C*).

## Appendix 5

### Two-component model of catalytic growth

Centrosome maturation involves many proteins but decades of studies have uncovered the essential molecular players whose interactions constitute a general motif of centrosome growth conserved across various organisms (main text *Table 2*). We consider a *two-component growth model* where the centrosome grows via forming a PCM scaffold of two scaffold formers. The first scaffold former ($a$) can get incorporated by the centriole and the second scaffold former $b$ binds to the first scaffold former to form an intermediate $b_i$ that can disassemble fast from the PCM. This consideration is based on the experimentally observed interactions between two essential centrosome proteins Spd-2/SPD-2 (fly/worms) and Cnn/SPD-5 which correspond to $a$ and $b$ components respectively. These two proteins are known to induce a positive feedback on centrosome growth via a kinase Polo/PLK1. The above-described dynamics (of $a$ and $b$) has a positive feedback on $b$ from $a$ by construction. A direct positive feedback from $b$ on $a$ (without considering the kinase/enzyme explicitly), such that the assembly (disassembly) rate of $a$ increases (decreases) with $a$, will result in autocatalytic feedback in centrosome growth that will give rise to centrosome size inequality. Below, we discuss how a positive feedback in growth can be constructed via the kinase/enzyme activity using the two-component model.

### Localized enzyme activity results in centrosome size inequality

The size of a growing centrosome is represented by the size of the growing PCM scaffold. We consider the two scaffolds $S_n(a)$ and $S_n(b)$ to be interleaved and composed of $n$ incorporated subunits of components $a$ and $b$. Total centrosome size (volume) is given by $V = V_a + V_b = (S_n(a) + S_n(b))\delta v$ where $V_a$ and $V_b$ are the volumes of the two scaffolds. We denote the enzyme/kinase abundance as $E$, whose abundance in the active form is denoted by $E^*$. Recent studies in *Drosophila* report that Cnn is specifically phosphorylated at the centrosome by the Polo kinase which is activated in the Spd-2 scaffold during centrosome maturation *Conduit et al., 2014a*; *Alvarez Rodrigo et al., 2019*; *Conduit et al., 2014b*. First, we consider the case of localized activity of the enzyme. This localized activity is induced as the enzyme being activated by the $a$-scaffold ($S(a)$) will phosphorylate the other scaffold former $b$ in an intermediate form ($b_i$) within that centrosome. For instance, the enzyme activated in the $a$-scaffold ($S_n^1(a)$) of centrosome-1, $E_1^*$, will only phosphorylate the intermediate form of the other scaffold former ($b_i^1$) present in centrosome-1. The full set of reactions are provided in *Figure 6—figure supplement 1A*. The resulting dynamics for a pair of centrosomes show significant centrosome size inequality (*Figure 6—figure supplement 1B*), which amplifies with increasing initial size difference (*Figure 6—figure supplement 1C*). Thus, the localized activity of the enzyme creates an effective autocatalytic feedback which leads to this size inequality.

### Shared enzyme activity leads to robust size regulation

The Polo kinase in *Drosophila* centrosome has a much faster turnover rate than the scaffold former proteins Spd-2 and *Conduit et al., 2014b*; *Conduit et al., 2015a*; *Wong et al., 2022*. We thus hypothesize that the activated Polo may be released from the scaffold and form a cytoplasmic pool that is shared between the two centrosomes, thereby enhancing the rate of growth of both the centrosomes. We incorporate this by considering the enzyme activation by the $a$-scaffold to be de-localized, i.e., the enzyme $E$ can be activated in the $a$-scaffold of any of the two centrosomes and released in the pool as $E^*$. This shared active enzyme then can phosphorylate the other scaffold former in any of the two centrosomes and enhance the rate of incorporation of $b$ into the $b$-scaffold of that centrosome. For a detailed description with all the constitutive reactions, see *Figure 6—figure supplement 2A*. This growth mechanism does not exhibit any individual size dependent positive feedback and can achieve size regulation for a pair of centrosomes with the characteristic sigmoidal growth curve (*Figure 6—figure supplement 2B*). The size inequality is insignificant and iindependent of the initial size difference between the centrosomes, indicating a robust regulation of size (*Figure 6—figure supplement 2C*). We further explore the effect of the overall enzyme concentration in determining the size of the centrosome and find that the enzyme concentration can regulate the centrosome size at the end of the maturation process (*Figure 6—figure supplement 2D*), as has been reported in experiments *Ohta et al., 2021*. The model predicts that the enzyme availability can signal the beginning and the end of the centrosome maturation process and a continuous enzyme dynamics is required to maintain the centrosome size (*Figure 6—figure supplement 2E*), consistent with experimental reports *Mahen et al., 2011*.

