## [Editor Report · eLife Assessment]

This **valuable** work suggests a new physical model of centrosome maturation: a catalytic growth model with a shared enzyme pool. The authors provide **compelling** evidence to show that the model is able to reproduce various experimental results such as centrosome size scaling with cell size and centrosome growth curves in *C. elegans*, and that the final centrosome size is more robust to differences in initial centrosome size. While direct experimental support for this theory is currently lacking, the authors propose concrete experiments that could distinguish their shared-enzyme model from previously proposed alternatives.

---

## [Referee Report · Reviewer #1 (Public review)]

The work analyzes how centrosomes mature before cell division. A critical aspect is the accumulation of pericentriolar material (PCM) around the centrioles to build competent centrosomes that can organize the mitotic spindle. The present work builds on the idea that the accumulation of PCM is catalyzed either by the centrioles themselves (leading to a constant accumulation rate) or by enzymes activated by the PCM itself (leading to autocatalytic accumulation). These ideas are captured by a previous model derived for PCM accumulation in *C. elegans* (Zwicker et al, PNAS 2014) and are succinctly summarized by Eq. 1. The main addition of the present work is to allow the activated enzymes to diffuse in the cell, so they can also catalyze the accumulation of PCM in other centrosomes (captured by Eqs. 2-4). The authors show that this helps centrosomes to reach the same size, independent of potential initial mismatches.

A strength of the paper is the simplicity of the equations, which are reduced to the bare minimum and thus allow a detailed inspection of the physical mechanism, e.g., using linear stability analysis. The possible shortcoming of this approach, namely that all equations assume that the diffusion of molecules is much faster than any of the reactive time scales, is addressed in Appendix 4. The authors show convincingly that their model compensates for initial size differences in centrosomes and leads to more similar final sizes. They carefully discuss parameter values used in their model, and they propose concrete experiments to test the theory. The model could thus stimulate additional experiments and help us understand how cells tightly control their centrosomes, which is crucial for faithful mitosis.

Comments on revised version:

The authors addressed my comments satisfactorily.

---

## [Referee Report · Reviewer #2 (Public review)]

In this paper, Banerjee & Banerjee argue that a solely autocatalytic assembly model of the centrosome leads to size inequality. The authors instead propose a catalytic growth model with a shared enzyme pool. Using this model, the authors predict that size control is enzyme-mediate and are able to reproduce various experimental results such as centrosome size scaling with cell size and centrosome growth curves in *C. elegans*.

The paper contains interesting results and is well-written and easy to follow/understand.

Comments on revised version:

The authors made a number of revisions that significantly improved the manuscript, including analyzing the impact of finite diffusion, more thorough stability analysis, and enhanced comparison to experimental results.

---

## [Author Response]

The following is the authors’ response to the original reviews.

**Public Reviews:**

**Reviewer #1 (Public Review):**
The work analyzes how centrosomes mature before cell division. A critical aspect is the accumulation of pericentriolar material (PCM) around the centrioles to build competent centrosomes that can organize the mitotic spindle. The present work builds on the idea that the accumulation of PCM is catalyzed either by the centrioles themselves (leading to a constant accumulation rate) or by enzymes activated by the PCM itself (leading to autocatalytic accumulation). These ideas are captured by a previous model derived for PCM accumulation in *C. elegans* (ref. 8) and are succinctly summarized by Eq. 1. The main addition of the present work is to allow the activated enzymes to diffuse in the cell, so they can also catalyze the accumulation of PCM in other centrosomes (captured by Eqs. 2-4). The authors claim that this helps centrosomes to reach the same size, independent of potential initial mismatches.A strength of the paper is the simplicity of the equations, which are reduced to the bare minimum and thus allow a detailed inspection of the physical mechanism. One shortcoming of this approach is that all equations assume that the diffusion of molecules is much faster than any of the reactive time scales, although there is no experimental evidence for this.

We appreciate the reviewer’s recognition of the strengths of our work. Indeed, the centrosome growth model incorporates multiple timescales corresponding to various reactions, and existing experimental data do not directly provide diffusion constants for the cytosolic proteins. However, we can estimate these diffusion constants using protein mass, based on the Stokes-Einstein relation, and compare the diffusion timescales with the reaction timescales obtained from FRAP analysis. For example, we estimate that the diffusion timescale for centrosomes separated by 5-10 micrometers is much smaller than the reaction timescales deduced from the FRAP experiments. Specifically, for SPD-5, a scaffold protein with a mass of ~150 kDa, the estimated diffusion constant is ~17 µm^2^/s, using the Stokes-Einstein relation and a reference diffusion constant of ~30 µm^2^/s for a 30 kDa GFP protein (reference: Bionumbers book). This results in a diffusion timescale of ~1 second for centrosomes 10 µm apart. In contrast, FRAP recovery timescales for SPD-5 in *C. elegans* embryos are on the order of several minutes, suggesting that scaffold protein binding reactions are much slower than diffusion. Therefore, a reaction-limited model is appropriate for studying PCM self-assembly during centrosome maturation. We have revised the manuscript to clarify this point and to include a discussion of the diffusion and reaction timescales.

Spatially extended model with diffusion

Both the reviewers have pointed out the importance of considering diffusion effects in centrosome size dynamics, and we agree that this is important to explore. We have developed a spatially extended 3D version of the centrosome growth model, incorporating stochastic reactions and diffusion (see Appendix 4). In this model, the system is divided into small reaction volumes (voxels), where reactions depend on local density, and diffusion is modeled as the transport of monomers/building blocks between voxels.

We find that diffusion can alter the timescales of growth, particularly when the diffusion timescale is comparable to or slower than the reaction timescale, potentially mitigating size inequality by slowing down autocatalysis. However, the main conclusions of the catalytic growth model remain unchanged, showing robust size regulation independent of diffusion constant or centrosome separation (Figure 2—figure supplement 3). Hence, we focused on the effect of subunit diffusion on the autocatalytic growth model. We find that in the presence of diffusion, the size inequality reduces with increasing diffusion timescale, i.e., increasing distance between centrosomes and decreasing diffusion constant (Figure 2—figure supplement 4). However, the lack of robustness in size control in the autocatalyic growth model remains, i.e., the final size difference increases with increasing initial size difference. Notably, in the diffusion-limited regime (very small diffusion or large distances), the growth curve loses its sigmoidal shape, resembling the behavior in the non-autocatalytic limit (Figure 2). These findings are discussed in the revised manuscript.

Another shortcoming of the paper is that it is not clear what species the authors are investigating and how general the model is. There are huge differences in centrosome maturation and the involved proteins between species. However, this is not mentioned in the abstract or introduction. Moreover, in the main body of the paper, the authors mention *C. elegans* on pages 2 and 3, but refer to Drosophila on page 4, switching back to *C. elegans* on page 5, and discuss Drosophila on page 6. This is confusing and looks as if they are cherry-picking elements from various species. The original model in ref. 8 was constructed for *C. elegans* and it is not clear whether the autocatalytic model is more general than that. In any case, a more thorough discussion of experimental evidence would be helpful.

We believe one strength of our approach is its applicability across organisms. Our goal in comparing the theoretical model with experimental data from *C. elegans* and *D.*

*melanogaster* is to demonstrate that the apparent qualitative differences in centrosome growth across species (see e.g., the extent of size scaling discussed in the section “Cytoplasmic pool depletion regulates centrosome size scaling with cell size”) may arise from the same underlying mechanisms in the theoretical model, albeit with different parameter values. We acknowledge differences in regulatory molecules between species, but the core pathways remain conserved see e.g. Raff, *Trends in Cell Biology* 2019, section: “Molecular Components of the Mitotic Centrosome Scaffold Appear to Have Been Conserved in Evolution from Worms to Humans”. In the revised manuscript, we have expanded the introduction to clarify this point and explain how our theory applies across species. We have also provided a clearer discussion of the experimental systems used throughout the manuscript and the available experimental evidence.

The authors show convincingly that their model compensates for initial size differences in centrosomes and leads to more similar final sizes. These conclusions rely on numerical simulations, but it is not clear how the parameters listed in Table 1 were chosen and whether they are representative of the real situation. Since all presented models have many parameters, a detailed discussion on how the values were picked is indispensable. Without such a discussion, it is not clear how realistic the drawn conclusions are. Some of this could have been alleviated using a linear stability analysis of the ordinary differential equations from which one could have gotten insight into how the physical parameters affect the tendency to produce equal-sized centrosomes.

Following the suggestion of the reviewer, we have revised the manuscript to add references and discussions justifying the choice of the parameter values used for the numerical simulations. These references and parameter choices can be found in Table 1 and Table 2, and are also discussed in relevant figure captions and within the manuscript text.

We thank the reviewer for the excellent suggestion of including linear stability analysis of the ODE models of centrosome growth. We included linear stability analyses of the catalytic and autocatalytic growth models in Appendix 3. Analysis of the catalytic growth model reaffirms the robustness of size equality and the analysis of autocatalytic growth provides an approximate condition of size inequality. We have modified the revised manuscript to discuss these results.

The authors use the fact that their model stabilizes centrosome size to argue that their model is superior to the previously published one, but I think that this conclusion is not necessarily justified by the presented data. The authors claim that "[...] none of the existing quantitative models can account for robustness in centrosome size equality in the presence of positive feedback." (page 1; similar sentence on page 2). This is not shown convincingly. In fact, ref 8. already addresses this problem (see Fig. 5 in ref. 8) to some extent.

The linear stability analysis shown in Fig 5 in ref 8 (Zwicker et al, PNAS, 2014) shows that the solutions are stable around the fixed point and it was inferred from this result that Ostwald ripening can be suppressed by the catalytic activity of the centriole, therefore stabilizing the centrosomes (droplets) against coarsening by Ostwald ripening. But, if size discrepancy arises from the growth process (e.g., due to autocatalysis) the timescale of relaxation for such discrepancy is not clear from the above-mentioned result. We show (in figure 2 - figure supplement 3) that for any appreciable amount of positive feedback, the solution moves very slowly around the fixed point (almost like a line attractor) and cannot reach the fixed point in a biologically relevant timescale. Hence the model in ref 8 does not provide a robust mechanism for size control in the presence of autocatalytic growth. We have added this discussion in the Discussion section.

More importantly, the conclusion seems to largely be based on the analysis shown in Fig. 2A, but the parameters going into this figure are not clear (see the previous paragraph). In particular, the initial size discrepancy of 0.1 µm^3 seems quite large, since it translates to a sphere of a radius of 300 nm. A similarly large initial discrepancy is used on page 3 without any justification. Since the original model itself already showed size stability, a careful quantitative comparison would be necessary.

We thank the reviewer for the valuable suggestions. The parameters used in Fig. 2A are listed in Table 1 with corresponding references, and we used the parameter values from Zwicker et al. (2014) for rate constants and concentrations.

The issue of initial size differences between centrosomes is important, but quantitative data on this are not readily available for *C. elegans* and *Drosophila*. Centrosomes may differ initially due to disparities in the amount and incorporation rate of PCM between the mother and daughter centrioles. Based on available images and videos (Cabral et al, Dev. Cell, 2019, DOI: https://doi.org/10.1016/j.devcel.2019.06.004), we estimated an initial radius of ~0.5 μm for centrosomes. Accounting for a 5% radius difference would lead to a volume difference of ~0.1 μm^3^, which was used in our analysis (Fig. 2A). These differences likely arise from distinct growth conditions of centrosomes containing different centrioles (older mother and newer daughter).

More importantly, we emphasize that the initial size difference does not qualitatively alter the results presented in Figure 2. We agree that a quantitative analysis will further clarify our conclusions, and we have revised the manuscript accordingly. For example, Figure 2—figure supplement 3 provides a detailed analysis of how the final centrosome size depends on initial size differences across various parameter values. Additionally, Appendix 3 now includes analytical estimates of the onset of size inequality as a function of these parameters.

The analysis of the size discrepancy relies on stochastic simulations (e.g., mentioned on pages 2 and 4), but all presented equations are deterministic. It's unclear what assumptions go into these stochastic equations, and how they are analyzed or simulated. Most importantly, the noise strength (presumably linked to the number of components) needs to be mentioned. How is this noise strength determined? What are the arguments for this choice? This is particularly crucial since the authors quote quantitative results (e.g., "a negligible difference in steady-state size (∼ 2% of mean size)" on page 4).

As described in the Methods, we used the exact Gillespie method (Gillespie, JPC, 1977) to simulate the evolution of the stochastic trajectories of the systems, corresponding to the deterministic growth and reaction kinetics outlined in the manuscript. We've expanded the Methods to include further details on the stochastic simulations and refer to Appendix 1, where we describe the chemical master equations governing autocatalytic growth..

The noise strength (fluctuations about the mean size of centrosome) does depend on the total monomer concentration (the pool size), and this may affect size inequality. Similar values of the total monomer concentration were used in the catalytic (0.04 uM) and autocatalytic growth (0.33 uM) simulations. These values for the pool size are similar to previous studies (Zwicker et al, PNAS, 2012) and have been optimized to obtain a good fit with experimental growth curves from *C. elegans* embryo data.

To present more quantitative results, we have revised our manuscript to add data showing the effect of pool size on centrosome size inequality (Figure 3 - figure supplement 2). We find the size inequality in catalytic growth to increase with decreasing pool size as the origin of this inequality is the stochastic fluctuation in individual centrosome size. The size inequality (ratio of dv/) in the autocatalytic growth does not depend (strongly) on the pool size (dv and both increase similarly with pool size).

Moreover, the two sets of testable predictions that are offered at the end of the paper are not very illuminative: The first set of predictions, namely that the model would anticipate an "increase in centrosome size with increasing enzyme concentration, the ability to modify the shape of the sigmoidal growth curve, and the manipulation of centrosome size scaling patterns by perturbing growth rate constants or enzyme concentrations.", are so general that they apply to all models describing centrosome growth. Consequently, these observations do not set the shared enzyme pool apart and are thus not useful to discriminate between models. The second part of the first set of predictions about shifting "size scaling" is potentially more interesting, although I could not discern whether "size scaling" referred to scaling with cell size, total amount of material, or enzymatic activity at the centrioles. The second prediction is potentially also interesting and could be checked directly by analyzing published data of the original model (see Fig. 5 of ref. 8). It is unclear to me why the authors did not attempt this.

In response to the reviewers' valuable feedback, we have revised the manuscript to include results on potential methods for distinguishing catalytic growth from autocatalytic growth. Since the growth dynamics of a single centrosome do not significantly differ between these two models, it is necessary to experimentally examine the growth dynamics of a centrosome pair under various initial size perturbations. In Figure 3-figure supplement 2, we present theoretical predictions for both catalytic and autocatalytic growth models, illustrating the correlation between initial and final sizes after maturation. The figure demonstrates that the initial size difference and final size difference should be correlated only in the autocatalytic growth and the relative size inequality decreases with increasing subunit pool size in catalytic growth while remains almost unchanged in autocatalytic growth. These predictions can be experimentally examined by inducing varying centrosome sizes at the early stage of maturation for different expression levels of the scaffold former proteins.

A second experimentally testable feature of the catalytic growth model involves sharing of the enzyme between both centrosomes. This could be tested through immunofluorescent staining of the kinase or by constructing a FRET reporter for PLK1 activity, where it can be studied if the active form of the PLK1 is found in the cytoplasm around the centrosomes indicating a shared pool of active enzyme. Additionally, photoactivated localization microscopy could be employed, where fluorescently tagged enzyme can be selectively photoactivated in one centrosome and intensity can be measured at the other centrosome to find the extent of enzyme sharing between the centrosomes.

We also discuss shifts in centrosome size scaling behavior with cell size by varying parameters of the catalytic growth model (Fig 4). While quantitative analysis of size scaling in *Drosophila* is currently unavailable, such an investigation could enable us to distinguish catalytic growth mode with other models. We have included this point in the Discussion section.

“The second prediction is potentially also interesting …” We assume the reviewer is referencing the scenario in Zwicker et al. (ref 8), where differences in centriole activity lead to unequal centrosome sizes. The data in that study represent a case of centrosome growth with variable centriole activity, resulting in size differences in both autocatalytic and catalytic growth models. This differs from our proposed experiment, where we induce unequal centrosome sizes without modifying centriole activity. We have now revised the text to clarify this distinction.

Taken together, I think the shared enzyme pool is an interesting idea, but the experimental evidence for it is currently lacking. Moreover, the model seems to make little testable predictions that differ from previous models.

We appreciate the reviewer’s interest in the core idea of our work. As mentioned earlier, we have improved the clarity in model predictions in the revised discussion section. Unfortunately, the lack of publicly available experimental data limits our ability to provide more direct experimental evidence. However, we are hopeful that our theoretical model will inspire future experiments to test these model predictions.

**Reviewer #2 (Public Review):**
Summary:In this paper, Banerjee & Banerjee argue that a solely autocatalytic assembly model of the centrosome leads to size inequality. The authors instead propose a catalytic growth model with a shared enzyme pool. Using this model, the authors predict that size control is enzyme-mediate and are able to reproduce various experimental results such as centrosome size scaling with cell size and centrosome growth curves in *C. elegans*.The paper contains interesting results and is well-written and easy to follow/understand.

We are delighted that the reviewer finds our work interesting, and we appreciate the thoughtful suggestions provided. In response, we have revised the text and figures to incorporate these recommendations. Below, we address each of the reviewer’s comments point by point:

Suggestions:● In the Introduction, when the authors mention that their "theory is based on recent experiments uncovering the interactions of the molecular components of centrosome assembly" it would be useful to mention what particular interactions these are.

As the reviewer suggested, we have modified the introduction section to add the experimental observations upon which we build our model.

● In the Results and Discussion sections, the authors note various similarities and differences between what is known regarding centrosome formation in C. elegan and Drosophila. It would have been helpful to already make such distinctions in the Introduction (where some phenomena that may be *C. elegans* specific are implied to hold centrosomes universally). It would also be helpful to include more comments for the possible implications for other systems in which centrosomes have been studied, such as human, Zebrafish, and Xenopus.

We thank the reviewer for this suggestion. We have modified the Introduction to motivate the comparative study of centrosome growth in different organisms and draw relevant connections to centrosome growth in other commonly studied organisms like Zebrafish and .

● For Fig 1.C, the two axes are very close to being the same but are not. It makes the graph a little bit more difficult to interpret than if they were actually the same or distinctly different. It would be more useful to have them on the same scale and just have a legend.

We have modified the Figure 1C in the revised manuscript. The plot now shows the growth of a single and a pair of centrosomes both on the same y-axis scale.

● The authors refer to Equation 1 as resulting from an "active liquid-liquid phase separation", but it is unclear what that means in this context because the rheology of the centrosome does not appear to be relevant.

We used the term “active liquid-liquid phase separation” simply to refer to a previous model proposed by Zwicker et al (PNAS, 2014) where the underlying process of growth results from liquid-liquid phase separation. We agree with the reviewer that the rheological property of the centrosome is not very relevant in our discussions and we have thus removed the sentence from the revised manuscript to avoid any confusion.

● The authors reject the non-cooperative limit of Eq 1 because, even though it leads to size control, it does not give sigmoidal dynamics (Figure 2B). While I appreciate that this is just meant to be illustrative, I still find it to be a weak argument because I would guess a number of different minor tweaks to the model might keep size control while inducing sigmoidal dynamics, such as size-dependent addition of loss rates (which could be due to reactions happen on the surface of the centrosome instead of in its bulk, for example). Is my intuition incorrect? Is there an alternative reason to reject such possible modifications?

The reviewer raises an interesting point here. However, we disagree with the idea that minor adjustments to the model can produce sigmoidal growth curves while still maintaining size control. In the absence of an external, time-dependent increase in building block concentration (which would lead to an increasing growth rate), achieving sigmoidal growth requires a positive feedback mechanism in the growth rate. This positive feedback alone could introduce size inequality unless shared equally between the centrosomes, as it is in our model of catalytic growth in a shared enzyme pool. The proposed modification involving size-dependent addition or loss rates due to surface assembly/disassembly may result in unequal sizes precisely because of this positive feedback. A similar example is provided in Appendix 1, where assembly and disassembly across the pericentriolic material volume lead to sigmoidal growth but also generate significant size inequality and lack of robustness in size control.

● While the inset of Figure 3D is visually convincing, it would be good to include a statistical test for completeness.

Following the reviewer’s suggestion, we present a statistical analysis in Figure 3 - Figure supplement 2 in the modified manuscript to enhance clarity. We show that the size difference values are uncorrelated (Pearson’s correlation coefficient ~ 0) with the initial size difference indicating the robustness of the size regulation mechanism.

● The authors note that the pulse in active enzyme in their model is reminiscent of the Polo kinase pulse observed in Drosophila. Can the authors use these published experimental results to more tightly constrain what parameter regime in their model would be relevant for Drosophila? Can the authors make predictions of how this pulse might vary in other systems such as *C. elegans*?

Thank you for the insightful suggestion regarding the use of pulse dynamics in experiments to better constrain the model’s parameter regime. In our revised manuscript, we attempted this analysis; however, the data from Wong et al. (EMBO 2022) for *Drosophila* are presented as normalized intensity in arbitrary units, rather than as quantitative measures of centrosome size or Polo enzyme concentration. This lack of quantitative data limits our ability to benchmark the model beyond capturing qualitative trends. We thus believe that quantitative measurements of centrosome size and enzyme concentration are necessary to achieve a tighter alignment between model predictions and biological data.

We discuss the enzyme dynamics in *C. elegans* in the revised manuscript. We find the enzyme dynamics corresponding to the fitted growth curves of *C. elegans* centrosomes are distinctly different from the ones observed in *Drosophila*. Instead of the pulse-like feature, we find a step-like increase in (cytosolic) active enzyme concentration.

● The authors mention that the shared enzyme pool is likely not diffusion-limited in *C. elegans* embryos, but this might change in larger embryos such as Drosophila or Xenopus. It would be interesting for the authors to include a more in-depth discussion of when diffusion will or will not matter, and what the consequence of being in a diffusion-limit regime might be.

Both the reviewers have pointed out the importance of considering diffusion effects in centrosome size dynamics, and we agree that this is important to explore. We have developed a spatially extended 3D version of the centrosome growth model, incorporating stochastic reactions and diffusion (see Appendix 4). In this model, the system is divided into small reaction volumes (voxels), where reactions depend on local density, and diffusion is modeled as the transport of monomers/building blocks between voxels.

We find that diffusion can alter the timescales of growth, particularly when the diffusion timescale is comparable to or slower than the reaction timescale, potentially mitigating size inequality by slowing down autocatalysis. However, the main conclusions of the catalytic growth model remain unchanged, showing robust size regulation independent of diffusion constant or centrosome separation (Figure 2—figure supplement 3). Hence, we focused on the effect of subunit diffusion on the autocatalytic growth model. We find that in the presence of diffusion, the size inequality reduces with increasing diffusion timescale, i.e., increasing distance between centrosomes and decreasing diffusion constant (Figure 2—figure supplement 4). However, the lack of robustness in size control in the autocatalyic growth model remains, i.e., the final size difference increases with increasing initial size difference. Notably, in the diffusion-limited regime (very small diffusion or large distances), the growth curve loses its sigmoidal shape, resembling the behavior in the non-autocatalytic limit (Figure 2). These findings are discussed in the revised manuscript.

● The authors state "Firstly, our model posits the sharing of the enzyme between both centrosomes. This hypothesis can potentially be experimentally tested through immunofluorescent staining of the kinase or by constructing FRET reporter of PLK1 activity." I don't understand how such experiments would be helpful for determining if enzymes are shared between the two centrosomes. It would be helpful for the authors to elaborate.

Our results indicate the necessity of the centrosome-activated enzyme to be shared for the robust regulation of centrosome size equality. If a FRET reporter of the active form of the enzyme (e.g., PLK1) can be constructed then the localization of the active form of the enzyme may be determined in the cytosol. We propose this based on reports of studying PLK activities in subcellular compartments using FRET as described in Allen & Zhang, BBRC (2006). Such experiments will be a direct proof of the shared enzyme pool. Following the reviewer’s suggestion, we have modified the description of the FRET based possible experimental test for the shared enzyme pool hypothesis in the revised manuscript.

Additionally, we have added another possible experimental test based on photoactivated localization microscopy (PALM), where tagged enzyme can be selectively photoactivated in one centrosome and intensity measured at the other centrosome to indicate whether the enzyme is shared between the centrosomes.

**Recommendations for the authors:**
The manuscript needs to clarify better what species the model describes, how alternative models were rejected, and how the parameters were chosen.

In the revised manuscript, we have connect the chemical species in our model to those documented in organisms like *Drosophila* and *C. elegans*. This connection is detailed in the main text under the Catalytic Growth Model section and summarized in Table 2. We discuss alternative models and our reasons for excluding them in the first results section on autocatalytic growth, with additional details provided in Appendix 1 and the accompanying supplementary figures. The selection of model parameters is addressed in the main text and methods, with references listed in Table 1. We believe that these revisions, along with our point-by-point responses to reviewer comments, comprehensively address all reviewer concerns.

**Reviewer #1 (Recommendations For The Authors):**
I think the style and structure of the paper could be improved on at least two accounts:(1) What's the role of the last section ("Multi-component centrosome model reveals the utility of shared catalysis on centrosome size control.")? It seems to simply add another component, keeping the essential structure of the model untouched. Not surprisingly, the qualitative features of the model are preserved and quantitative features are not discussed anyway.

This model provides a more realistic description of centrosome growth by incorporating the dynamics of the two primary scaffold-forming subunits and their interactions with an enzyme. It is based on the observation that the major interaction pathways among centrosome components are conserved across many organisms (see Raff, *Trends in Cell Biology*, 2019 and Table 2), typically involving two scaffold-forming proteins and one enzyme that mediates positive feedback between them. These pathways may involve homologous proteins in different species.

This model allows us to validate the experimentally observed spatial spread of the two subunits, Cnn and Spd-2, in *Drosophila*. Additionally, we used it to investigate the impact of relaxing the assumption of a shared enzyme pool on size control. Although similar insights could be obtained using a single-component model, the two-component model offers a more biologically relevant framework. We have highlighted these points in the revised manuscript to ensure clarity.

(2) The very long discussion section is not very helpful. First, it mostly reiterates points already made in the main text. Second, it makes arguments for the choice of modeling (top left column of page 8), which probably should have been made when introducing the model. Third, it introduces new results (lower left column of page 8), which should probably be moved to the main text. Fourth, the interpretation of the model in light of the known biochemistry is useful and should probably be expanded although I think it would be crucial to keep information from different organisms clearly separate (this last point actually holds for the entire manuscript).

We thank the reviewer for the feedback. We have modified the discussion section to focus more on the interpretation of the results, model predictions and future outlook with possible experiments to validate crucial aspects of the model. We have moved most of the justifications to the main text model description.

Here are a few additional minor points:* page 1: Typo "for for" → "for"* Page 8: Typo "to to" → "to"

We thank the reviewer for the useful recommendations. We have corrected all the typos in the revised manuscript.

* Why can diffusion be neglected in Eq. 1? This is discussed only very vaguely in the main text (on page 3). Strangely, there is some discussion of this crucial initial step in the discussion section, although the diffusion time of PLK1 is compared to the centrosome growth time there and not the more relevant enzyme-mediate conversion rate or enzyme deactivation rate.

We now discuss the justification of neglecting diffusion while motivating the model. We have added a more detailed discussion in the Methods section. We estimate the timescale of diffusion for the scaffold formers and the enzyme and compare them with the turnover timescales of the respective proteins Spd-2, Cnn and Polo. We find the proteins to diffuse fast compared to their FRAP recovery timescales indicating reaction timescales to be slower than the timescales of diffusion. Nevertheless, following the reviewer’s suggestion, we have also investigated the effect of diffusion on the growth process in Appendix 4.

* Page 3: The comparison k_0^+ ≫ k_1^+ is meaningless without specifying the number of subunits n. I even doubt that this condition is the correct one since even if k_0^+ is two orders of magnitude larger than k_1^+, the autocatalytic term can dominate if there are many subunits.

We thank the reviewer for the insightful comment on the comparison between the growth rates k^+_0 and k^+_1. Indeed, the pool size matters and we have now included a linear stability analysis of the autocatalytic growth equations in Appendix 3 to estimate the condition for size inequality. We have commented on these new findings in the revised manuscript.

* The Eqs. 2-4 are difficult to follow in my mind. For instance, it is not clear why the variables N_av and N_av^E are introduced when they evidently are equivalent to S_1 and E. It would also help to explicitly mention that V_c is the cell volume. Moreover, do these equations contain any centriolar activity? If so, I could not understand what term mediates this. If not, it might be good to mention this explicitly.

Following the reviewer’s suggestion, we have modified the equations 2-4 and added the definition of V_c to enhance clarity in the revised manuscript. The centriole activity is given by k^+ in the catalytic model. We now explicitly mention it.

* Page 4: The observed peak of active enzyme (Fig 3C) is compared to experimental observation of a PLK1 peak at centrosomes in *Drosophila* (ref. 28). However, if I understand correctly, the peak in the model refers to active enzyme in the entire cell (and the point of the model is that this enzymatic pool is shared everywhere), whereas the experimental measurement quantified the amount of PLK1 at the centrosome (and not the activity of the enzyme). How are the quantity in the model related to the experimental measurements?

The reviewer is correct in pointing out the difference between the quantities calculated from our model and those measured in the experiment by Wong et al. We have clarified this point in the revised manuscript. We hypothesize that if, in future experiments, the active (phosphorylated) polo can be observed by using a possible FRET reporter of activity then the cytosolic pulse can be observed too. We discuss this point in the revised manuscript.

* Page 6: The asymmetry due to differences in centriolar activity is apparently been done for both models (Eq. 1 and Eqs. 2-4), referring to a parameter k_0^+ in both cases. How does this parameter enter in the latter model? More generally, I don't really understand the difference in the two rows in Fig. 5 - is the top row referring to growth driven by centriolar activity while the lower row refers to pure autocatalytic growth? If so, what about the hybrid model where both mechanisms enter? This is particularly relevant, since ref. 8 claims that such a hybrid model explains growth curves of asymmetric centrosomes quantitatively. Along these lines, the analysis of asymmetric growth is quite vague and at most qualitative. Can the models also explain differential growth quantitatively?

We believe the reviewer’s comment on centrosome size asymmetry may stem from a lack of clarity in our initial explanation. In this section, as shown in Figure 5, we compare the full autocatalytic model (where both k_0^+ and k_1^+ are non-zero) with the catalytic model. The confusion might have arisen due to an unclear definition of centriolar activity in the catalytic growth model, which we have clarified in the revised manuscript. Specifically, we use k+ in the catalytic model and k0+ in the autocatalytic model as indicators of centriolar activity.

Our findings quantitatively demonstrate that variations in centriole activity can robustly drive size asymmetry in catalytic growth, independent of initial size differences. However, in autocatalytic growth, increased initial size differences make the system more vulnerable to a loss of regulation, as positive feedback can amplify these differences, ultimately influencing the final size asymmetry. Our results do not contradict Zwicker et al. (ref 8); rather, they complement it. We show that size asymmetry in autocatalytic growth is governed by both centriole activity and positive feedback, highlighting that centriole activity alone cannot robustly regulate centrosome size asymmetry within this framework.

* The code for performing the simulations does not seem to be available

We have now made the main codes available in a GitHub repository. Link: https://github.com/BanerjeeLab/Centrosome_growth_model